# LogicTree: Improving Complex Reasoning of LLMs via Instantiated Multi-step Synthetic Logical Data

**Zehao Wang**[1,*], **Lin Yang**[2], **Jie Wang**[1,†], **Kehan Wang**[1], **Hanzhu Chen**[1], **Bin Wang**[2]
**Jianye Hao**[2,3], **Defu Lian**[1], **Bin Li**[1], **Enhong Chen**[1]
[1]MoE Key Laboratory of Brain Science and Education, Psychological and Cognition,
University of Science and Technology of China
[2]Noah's Ark Lab, Huawei Technologies, [3] Tianjin University

## Abstract

Despite their remarkable performance on various tasks, Large Language Models (LLMs) still struggle with logical reasoning, particularly in complex and multi-step reasoning processes. Among various efforts to enhance LLMs' reasoning capabilities, synthesizing large-scale, high-quality logical reasoning datasets has emerged as a promising direction. However, existing methods often rely on pre-defined templates for logical reasoning data generation, limiting their adaptability to real-world scenarios. To address the limitation, we propose **LogicTree**, a novel framework for efficiently synthesizing multi-step logical reasoning dataset that excels in both complexity and instantiation. By iteratively searching for applicable logic rules based on structural pattern matching to perform backward deduction, **LogicTree** constructs multi-step logic trees that capture complex reasoning patterns. Furthermore, we employ a two-stage LLM-based approach to instantiate various real-world scenarios for each logic tree, generating consistent real-world reasoning processes that carry contextual significance. This helps LLMs develop generalizable logical reasoning abilities across diverse scenarios rather than merely memorizing templates. Experiments on multiple benchmarks demonstrate that our approach achieves an average improvement of 9.4% in accuracy for LLMs on complex logical reasoning tasks.

## 1 Introduction

Logical reasoning is a crucial capability for large language models (LLMs) [15, 24, 14], providing substantial support for complex tasks such as coding, mathematics, and other higher-order cognitive abilities [30, 28, 45]. Recent advancements in the reasoning capabilities of LLMs have been impressive[46], with one of the key progress attributed to the availability of high-quality reasoning data[18, 9, 50, 6]. Despite these strides, LLMs still encounter challenges in complex multi-step logical reasoning tasks[4, 31, 42], underscoring a significant scarcity of high-quality multi-step logical reasoning datasets.

Early logical reasoning datasets are primarily constructed through manual data collection and annotation [19, 55, 13]. While these datasets are often diverse and intricate, their creation is labor-intensive and time-consuming for further training. In recent years, several studies have explored data synthesis approaches to generate logical reasoning data [37]. Some studies have employed a set of atomic logic rules to evaluate and enhance the formal reasoning abilities of LLMs[41, 33]. Similarly, other research has synthetically created multi-step logical reasoning datasets by leveraging natural language templates[30, 29, 7, 2]. Furthermore, there is an increasing interest in generating reasoning processes

---

[*]This word was done when Zehao Wang interned at Huawei. Email: zh-wang@mail.ustc.edu.cn
[†]Corresponding author. Email:jiewangx@ustc.edu.cn

39th Conference on Neural Information Processing Systems (NeurIPS 2025).

for existing complex logical reasoning questions[21, 39]. However, existing data synthesis methods typically exhibit the following limitations: (1) insufficient complexity, as they typically generate simplistic reasoning patterns with limited types of reasoning rules and shallow reasoning steps; and (2) inadequate real-world scenario instantiation, often resulting from the combination of entities that lack contextual or conceptual relevance, which has been shown to weaken the robustness and generalization capabilities of LLMs in reasoning tasks[51, 47]. Real-world scenario instantiation refers to the process of grounding synthetic reasoning data in concrete and context-rich real-world scenarios that incorporate semantically relevant entities or events, capable of reflecting the complexity and diversity of real-world reasoning tasks.

To address these challenges, we propose **LogicTree**, a novel framework for efficiently synthesizing multi-step logical reasoning dataset, offering significant advantages in terms of **complexity** and **instantiation**. Our method leverages first-order logic rules to generate complex multi-step logical reasoning trees, which are then instantiated using LLMs to produce natural language reasoning questions with real-world scenarios and reasoning processes that carry contextual significance. The logic trees generated by **LogicTree** exhibit complex reasoning patterns (as presented in Tables 10 and 11), facilitating the improvement of advanced reasoning abilities in LLMs for tackling complex tasks (Sec.4.3). Moreover, contextually diverse instantiated reasoning data help LLMs develop generalizable logical reasoning skills, rather than simply memorizing implicit relationships between facts[24, 51, 40].

Figure 1 presents an overview of **LogicTree**, illustrating the data synthesis process in detail. Firstly, we construct symbolic logical reasoning trees via a backward deduction procedure, where leaf nodes are iteratively expanded by applying formal logic rules identified through structural pattern matching of formulas. In this way, we generate multi-step logical reasoning trees that incorporate diverse and complex reasoning patterns, encompassing both propositional logic and first-order logic. Secondly, we introduce a **two-stage LLM-based approach** to instantiate logical reasoning trees: (1) assigning contextually relevant entities or events to the logical symbols in the leaf nodes to construct realistic reasoning scenarios; and (2) sequentially translating all intermediate nodes in the logical tree to generate step-by-step natural language reasoning processes. By controlling the thematic domains of the content generated by LLMs, we instantiate each logical reasoning tree with multiple diverse scenarios, thereby enhancing the overall diversity and contextual richness of the dataset (see Figures 4 and 3). Finally, we apply post-processing to the synthesized data to verify the logical consistency of the instantiated content and construct logical reasoning instances, each consisting of a set of premises, a conclusion–answer pair, and the corresponding reasoning process. Experiments conducted across a wide range of LLMs demonstrate that the instantiated complex logical reasoning data synthesized by LogicTree effectively enhances the models' reasoning capabilities. Furthermore, our analytical experiments (Table 3) indicate that diverse instantiation scenarios are beneficial for improving the generalization of reasoning abilities.

The main contributions of this work include:

- We introduce **LogicTree**, a novel framework for synthesizing multi-step logical reasoning datasets by generating complex logic trees with first-order logic rules and instantiating them into realistic reasoning scenarios, enabling LLMs to develop advanced and generalizable logical reasoning abilities (Sec.3).

- We present a **two-stage LLM-based instantiation technique** that injects realistic statements into the symbolic reasoning trees, generating coherent and contextually grounded natural language reasoning processes (Sec.3).

- Experimental results demonstrate that the synthesized dataset significantly enhance the logical reasoning performance of LLMs across several challenging benchmarks, leading to an average accuracy improvement of up to 9.4% (Sec.4).

## 2   Preliminary

Logical reasoning is a cognitive process that involves deriving valid conclusions from given premises based on formal **logic rules** [15, 27, 5]. It is a cornerstone of fields like artificial intelligence, mathematics, and philosophy, enabling systematic decision-making and problem-solving[24, 34]. The main formal logic systems include **propositional logic** and **first-order logic(FOL)**. Propositional

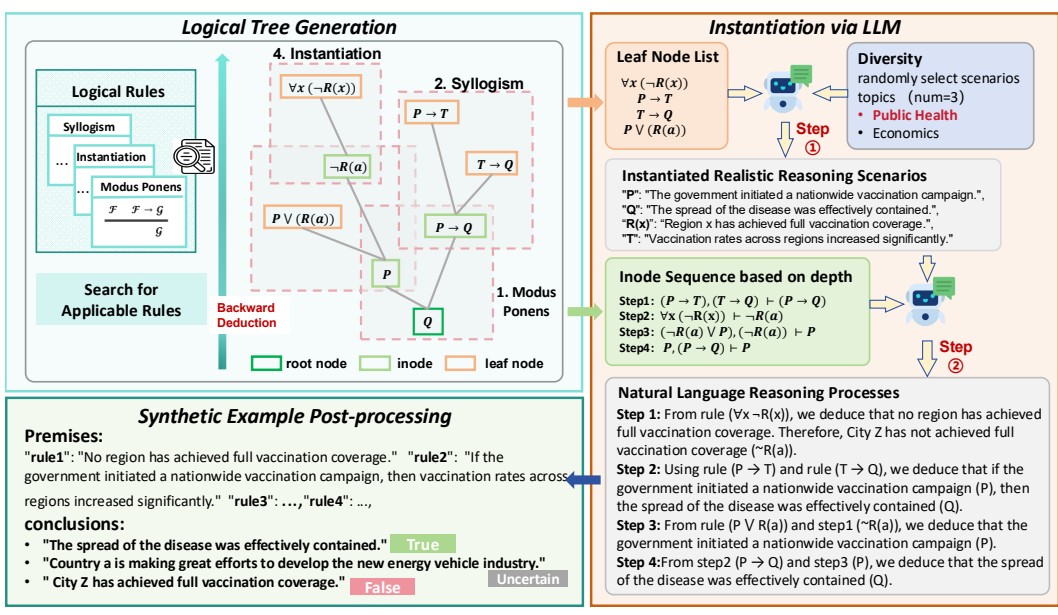

Figure 1: An overview of LogicTree, which comprises three key modules: (1) Logical reasoning tree generation via iterative backward deduction based on structural pattern matching; (2) Reasoning scenario instantiation using a two-stage LLM-based approach; (3) Synthetic reasoning example post-processing.

logic focuses on propositions or statements that are either true or false, employing logical operators such as ∧ (and), ∨ (or), ¬ (not), and → (implies) to connect these statements. In contrast, FOL extends propositional logic by handling more complex statements involving variables, quantifiers (e.g., ∀ for "for all" and ∃ for "there exists"), and predicates.

In a formal logic system, a logical expression composed of the aforementioned symbols is referred to as a **formula** (e.g., $A \rightarrow B$, $\forall x P(x)$). If an expression contains no logical operators, it is called an **atomic formula**. By representing premises or statements as formulas, logical rules can be systematically applied to derive new conclusions. For instance, *Universal Instantiation* and *Hypothetical Syllogism* are two well-known inference rules:

$$\text{UI:} \quad \forall x(p(x)) \vdash p(a) \tag{1}$$

$$\text{HS:} \quad \big((p \rightarrow q) \land (q \rightarrow r)\big) \vdash (p \rightarrow r) \tag{2}$$

Here, $\vdash$ denotes "derives", meaning that the formulas preceding it serve as premises, while the formula following it represents the conclusion. The *Hypothetical Syllogism (HS)* rule states that if "$p$ implies $q$" and "$q$ implies $r$", then we can conclude that "$p$ implies $r$". For additional logical rules and further details, please refer to Appendix C.

## 3   Method

In this section, we introduce the design and implementation of **LogicTree**, a novel framework for synthesizing instantiated complex logical reasoning datasets. Figure 1 overviews the workflow of our approach, which consists of three main modules: **logical reasoning tree generation**, **reasoning scenario instantiation** and **synthetic reasoning example post-processing**. First, based on structural pattern matching, we construct symbolic logical reasoning trees through a backward deduction process(Sec. 3.1). Next, we leverage LLMs to instantiate the logical reasoning trees with real-world scenarios(Sec.3.2). Finally, we verify the logical consistency of the instantiated content and generate synthetic logical reasoning instances(Sec.3.3).

### 3.1 Logical Reasoning Tree Generation

**Logical Reasoning Tree**   Each node in a logical reasoning tree (logic tree) corresponds to a logical expression (formula) that can be derived from its child nodes based on logic rules. The role of each node varies depending on its position within the tree: ***leaf nodes*** represent premises, the ***root node*** denotes the final conclusion, and ***intermediate nodes (inodes)*** encapsulate individual reasoning steps.

In order to construct logical reasoning tree,we designed a backward deduction method based on structural pattern matching of formulas, which starts from a given formula and searches for applicable logic rules to derive the premises that support it. When determining whether a logic rule can be applied to a formula, this method does not require the formula to be identical to the conclusion of the logic rule; instead, it only compares whether the structural patterns of their are matched. In this way, we generate multi-step logical reasoning trees that incorporate diverse and complex reasoning patterns, encompassing both propositional logic and first-order logic. We provide additional detailed information in the Appendix C.2, including the types of logical rules involved and the complexity of the logical trees (e.g., depth, breadth, and the number of rules).

---

**Algorithm 1** Logical Reasoning Tree Generation Process

---

1: **Input:** Root conclusion formula $\mathcal{Q}$, number of iterations $N$, rules list $\mathcal{R} = \{r_1, r_2, \ldots, r_m\}$
2: $\mathcal{T} = init\_logictree(\mathcal{Q})$
3: $k = 1$
4: **while** $k \leq N$ **do**
5: $\quad \mathcal{L} = random\_select\_leaf\_node(\mathcal{T})$
6: $\quad \mathcal{A} = \emptyset \quad$ *#appliable rules list*
7: $\quad$ **for** $r \in \mathcal{R}$ **do**
8: $\quad\quad$ **if** $is\_appliable\_rule(\mathcal{L}.AST, r.conclusion.AST)$ **then**
9: $\quad\quad\quad \mathcal{A} \leftarrow \mathcal{A} \cup \{r\}$
10: $\quad\quad$ **end if**
11: $\quad$ **end for**
12: $\quad appliable\_rule = random\_choice(\mathcal{A})$
13: $\quad \mathcal{L}.childs = backward\_deduction(\mathcal{L}, appliable\_rule)$
14: $\quad k = k + 1$
15: **end while**
16: **Output:** Final logical reasoning tree $\mathcal{T}$

---

**Structural Pattern Matching of Formulas**   Logical formulas can be represented as *Abstract Syntax Trees (ASTs)*. In this AST representation, the root node typically corresponds to the logical operator with the highest precedence. The operands of this operator are then represented by the subtrees branching from the root. This decomposition process is applied recursively to the subtrees, continuing until the tree's nodes represent only atomic formulas, which form the leaves of the tree. ***The structure of a formula is considered to match the structural pattern of another if their AST's respective operator skeletons are isomorphic.*** This means their ASTs have the same upper-level structure. For example, consider the formulas $(A \vee B) \rightarrow (\neg C)$ and $F \rightarrow G$. Although the two formulas are not identical, they have the same outermost logical operator, namely $\rightarrow$. Since $F \rightarrow G$ are the conclusions of Rule (2), we can apply this rule to $(A \vee B) \rightarrow (\neg C)$ for backward deduction, resulting in the premises $(A \vee B) \rightarrow G$ and $G \rightarrow (\neg C)$. Therefore, if the conclusion formula of a rule matches the structural pattern of the target formula, the rule can be applied to the target formula. Refer to Appendix C.1 and Figure 5 for more details.

**Multi-step Backward Deduction Generation**   As shown in Algorithm 1, our logical reasoning tree generation follows a backward deduction process, iteratively searching for applicable formal logical rules based on structural pattern matching to decompose existing formula nodes into new ones. Initially, the logical tree consists of a single root node, which is a randomly generated formula. Then, the process iterates as follows until the desired number of iterations is reached: Randomly select a leaf node in the tree, apply relevant rules to perform backward deduction, and derive the necessary premises, which are added as child nodes to the selected formula. The number of iterations and the search strategy can control the structure of the final logical reasoning tree, specifically its depth and width. Examples of logical reasoning trees can be found in the Appendix E.

## 3.2 Reasoning Scenario Instantiation via LLMs

Previous works[30, 29, 3] have employed template based approach to translate the symbolic formulas into natural language statements. However, these methods fail to capture the contextual semantics of logical relationships between formulas, often generating statements with limited real-world applicability.

Thus, we aim to instantiate logical reasoning trees with realistic reasoning scenarios, which involves translating symbolic logical expressions into concrete statements that carry contextual significance. Given the advanced text generation capabilities of modern LLMs (e.g., GPT-4 [1]), an intuitive approach is to leverage LLMs for instantiation. However, due to the inherent limitations of LLMs in handling complex logical reasoning, directly prompting them to generate realistic logical reasoning problems from symbolic logic trees may introduce errors. To mitigate this issue, we design the following two-state prompting strategy to guide LLMs in instantiating the logical reasoning tree. The details of prompts are provided in Appendix D.

(1) **Logical tree instantiation**: We extract all the formulas that are leaf nodes in the logical reasoning tree and input them into LLMs to instantiate a real-world reasoning scenario. During this process, LLMs assign real-world statements to atomic formulas while ensuring consistency with their predefined logical relationships. Once all leaf nodes are translated into natural language, a coherent and contextually grounded logical reasoning scenario is formed.

(2) **Reasoning process translation**: We obtain a complete, step-by-step symbolic reasoning chain by sorting the internal nodes of the logical tree in descending order of depth. At each step, an inode is derived from its child nodes according to the corresponding logical rules. We then leverage LLMs to translate the symbolic reasoning steps into a natural language reasoning process within the previously instantiated scenario.

In this process, **the LLM is guided by a correct and rigorous symbolic reasoning skeleton to generate a step-by-step, instantiated natural language reasoning process**. This approach significantly reduces generation errors, as the LLM is not required to perform the reasoning autonomously; it only needs to understand the logical relationships and perform symbolic translatio, which is well within the capabilities of LLMs[32, 36]. Furthermore, by **controlling the themes of the reasoning scenarios used by the LLM**, we can ensure diversity in the synthesized data (Figure 4). This approach enables the generation of coherent, step-by-step logical reasoning processes in natural language that incorporate a wide range of realistic scenarios.

## 3.3 Synthetic Reasoning Example Post-processing

We conduct a systematic verification of the logical consistency of the instantiated content to filter out erroneous data. Specifically, this involves prompting the LLM to output the instantiated natural language statements paired with their corresponding logical expressions, typically in the format "statement [expression]". We then perform an exact string comparison between the logical expression provided by the LLM and the original source logical expression to identify any inconsistencies. The entire natural language reasoning process is validated once all statements are confirmed to be logically consistent with their symbolic expressions, as the overall correctness is inherently guaranteed by the logic tree. This rigorous verification process enables us to filter out the data identified as erroneous, thereby significantly improving the overall data quality.

Then, we utilize the verified instantiated content to construct logical reasoning data instances. A logical reasoning instance is composed of a set of premises, a conclusion-answer pair, and the corresponding reasoning process. In particular, the natural language statements of the leaf nodes are combined to form the premises. Then we can construct conclusions with different types of answers ('True', 'False' or 'Uncertain') based on the logical tree. For answer label "True", we use the statement of root node as the conclusion. For answer label "False", we use the negated statement of the root node as the conclusion. For answer label "Uncertain", we introduce some irrelevant statements as distractors. Examples of synthetic logical reasoning data can be found in Appendix E.

# 4 Experiments

To evaluate the effectiveness of our **LogicTree**, we design a suite of experiments that not only demonstrate its significant impact on enhancing LLMs' reasoning abilities, but also provide in-depth analytical insights. Specifically, we divide the experiments into three parts:

- To comprehensively assess the enhancements of our approach in improving the logical reasoning abilities of LLMs, we conduct evaluations across multiple benchmarks and compare it against several other synthetic logical datasets (Sec.4.2).

- To further evaluate the model's multi-step logical reasoning ability, we assess **LogicTree** and baselines on Multi-LogiEval, which categorizes questions by reasoning steps from one to five (Sec.4.3).

- To analyze the key factors contributing to LogicTree's effectiveness, we conduct ablation studies on three core components: (1) instantiating real-world reasoning scenarios, (2) incorporating natural language reasoning processes under instantiated scenarios, and (3) ensuring diversity in instantiated scenarios (Sec.4.5).

- To further evaluate the generalization enhancement of model reasoning capabilities by LogicTree, we conduct additional experiments across a diverse range of task types, covering domains such as logic, math, code, NLI, and others (Sec.4.4).

- To assess the generalizability of our synthetic data, we extended our evaluation to include models from different families and scales, specifically Qwen2.5-1.5B/3B and the Deepseek-R1-Distill series (Appendix.B.3).

## 4.1 Experimental Setup

We briefly explain the experimental settings. Refer to Appendix B for more details.

**Syntheised Training Dataset.** We generated 5,000 symbolic logic trees with depths from 2 to 15, and instantiated each into 3 semantically diverse scenarios, yielding 15,000 reasoning problems. After applying an automatic filtering process that discarded 8.73% of noisy or invalid samples, the final dataset for LLM training contains 13.8k high-quality, multi-step reasoning instances.

**Evaluation Benchmarks.** To evaluate the effectiveness of our proposed **LogicTree**, we consider a diverse set of reasoning tasks and datasets that require rigorous logical reasoning: (a) **LogicBench**[33]: a novel task designed to comprehensively evaluate the model's performance on each inference rule. (b) **LogiQA2.0**[19]: A collection of challenging real-world logical reasoning problems from civil service entrance exams. (c) Three **BIG-Bench Hard (BBH)**[38] tasks of varying difficulty levels: logical deduction with three, five, and seven objects. (d) **FOLIO**[13]: An expert-written, logically complex dataset for first-order logic reasoning. (e) Two **AGIEval**[55] tasks focused on logical reasoning: LAST-AR and LAST-LR. (f) **Multi-LogiEval**[35]: A comprehensive dataset incorporating varying reasoning depths for logical complexity.

**Baselines.** To demonstrate the superiority of our method, we evaluated the following baselines. **(i) Vanilla**: standalone LLMs without additional training. Vanilla LLMs represent the original capabilities of LLMs. **(ii) PARARULE**[2]: it generates deductive processes by repeatedly applying deduction rules to a given set of facts and transforms these processes into sentences based on natural language templates. **(iii) LogicAsker**[41]: a framework to evaluate the ability of LLMs to handle individual atomic logical reasoning rules and generate targeted samples for improvement. **(iv) FLD**$_{\times 2}$[29]: it introduces systematic design principles for logical data synthesis and randomly combines various rules to construct multi-step reasoning datasets.

**Models and Training Settings.** To rigorously validate the effectiveness of LogicTree, we conduct extensive experiments using a diverse set of prominent open-source models spanning various families and scales. We utilize models from the Llama-3.1, Mistral-v0.3, Qwen2.5, and Deepseek-R1-Distill families, with parameter scales ranging from 1.5B to 70B. We employ two distinct fine-tuning strategies: full fine-tuning is applied to smaller-scale models (under 8B), while the larger 70B model utilizes LoRA fine-tuning.

Table 1: Main results on five types of reasoning tasks. $\triangle$ means the margin between Vanilla and training by LogicTree. "Avg" means the average accuracy across five benchmarks for each methods. We bold the best results for each LLM backbone and underline the second-best results.

| Model | LogicBench | LogiQA2.0 | FOLIO | BBH-Logic | AGIEval | | Avg. |
|---|---|---|---|---|---|---|---|
| | | | | | LR | AR | |
| **Llama-3.1-8B** | | | | | | | |
| vanilla | $80.0_{\pm0.4}$ | $42.4_{\pm0.2}$ | $50.8_{\pm0.4}$ | $39.3_{\pm0.3}$ | $48.4_{\pm0.3}$ | $20.7_{\pm0.1}$ | 46.9 |
| **+ PARARULE** | $84.4_{\pm0.3}$ | $53.3_{\pm0.2}$ | $50.3_{\pm0.3}$ | $42.4_{\pm0.1}$ | $\underline{52.9}_{\pm0.2}$ | $21.7_{\pm0.2}$ | 50.8 |
| **+ LogicAsker** | $\underline{85.0}_{\pm0.3}$ | $\mathbf{54.7}_{\pm0.2}$ | $\underline{54.6}_{\pm0.2}$ | $43.7_{\pm0.1}$ | $51.2_{\pm0.2}$ | $\underline{23.1}_{\pm0.1}$ | 52.0 |
| **+ FLD$_{\times2}$** | $83.6_{\pm0.2}$ | $53.3_{\pm0.2}$ | $54.5_{\pm0.3}$ | $39.5_{\pm0.2}$ | $51.8_{\pm0.3}$ | $21.3_{\pm0.1}$ | 50.7 |
| **+ LogicTree** | $\mathbf{90.6}_{\pm0.2}$ | $\underline{53.5}_{\pm0.1}$ | $\mathbf{57.6}_{\pm0.2}$ | $\mathbf{53.2}_{\pm0.2}$ | $\mathbf{56.1}_{\pm0.1}$ | $\mathbf{26.5}_{\pm0.2}$ | **56.3** |
| $\triangle$ **(Relative Gain)** | +10.6 (+13.3%) | +11.1 (+26.2%) | +6.8 (+13.4%) | +13.9 (+35.4%) | +7.7 (+15.9%) | +5.8 (+28.0%) | +9.4 (+20.0%) |
| **Qwen-2.5-7B** | | | | | | | |
| vanilla | $83.1_{\pm0.3}$ | $63.8_{\pm0.4}$ | $54.3_{\pm0.4}$ | $72.8_{\pm0.3}$ | $65.3_{\pm0.3}$ | $23.9_{\pm0.2}$ | 60.5 |
| **+ PARARULE** | $82.9_{\pm0.3}$ | $65.1_{\pm0.1}$ | $55.3_{\pm0.2}$ | $72.8_{\pm0.4}$ | $\underline{68.3}_{\pm0.2}$ | $24.5_{\pm0.1}$ | 61.5 |
| **+ LogicAsker** | $\underline{84.3}_{\pm0.3}$ | $\mathbf{64.9}_{\pm0.1}$ | $\underline{58.7}_{\pm0.2}$ | $72.8_{\pm0.2}$ | $66.7_{\pm0.2}$ | $\underline{24.5}_{\pm0.1}$ | 62.0 |
| **+ FLD$_{\times2}$** | $83.4_{\pm0.1}$ | $65.1_{\pm0.3}$ | $59.6_{\pm0.3}$ | $73.6_{\pm0.2}$ | $67.3_{\pm0.1}$ | $26.1_{\pm0.2}$ | 62.5 |
| **+ LogicTree** | $\mathbf{89.2}_{\pm0.1}$ | $\underline{64.9}_{\pm0.1}$ | $\mathbf{63.8}_{\pm0.1}$ | $\mathbf{74.3}_{\pm0.3}$ | $\mathbf{69.3}_{\pm0.1}$ | $\mathbf{28.7}_{\pm0.2}$ | **65.0** |
| $\triangle$ **(Relative Gain)** | +6.1 (+7.3%) | +1.1 (+1.7%) | +9.5 (+17.5%) | +1.5 (+2.1%) | +4.0 (+6.1%) | +4.8 (+20.1%) | +4.5 (+7.4%) |
| **Mistral-7B-v0.3** | | | | | | | |
| vanilla | $77.5_{\pm0.4}$ | $49.7_{\pm0.3}$ | $51.2_{\pm0.2}$ | $45.5_{\pm0.3}$ | $49.9_{\pm0.3}$ | $20.4_{\pm0.1}$ | 49.0 |
| **+ PARARULE** | $\underline{78.8}_{\pm0.2}$ | $\underline{51.2}_{\pm0.3}$ | $50.3_{\pm0.3}$ | $47.2_{\pm0.2}$ | $50.4_{\pm0.2}$ | $22.6_{\pm0.2}$ | 50.1 |
| **+ LogicAsker** | $78.8_{\pm0.3}$ | $50.6_{\pm0.2}$ | $\mathbf{56.7}_{\pm0.1}$ | $46.8_{\pm0.1}$ | $50.2_{\pm0.2}$ | $\underline{23.1}_{\pm0.1}$ | 51.0 |
| **+ FLD$_{\times2}$** | $78.0_{\pm0.2}$ | $50.6_{\pm0.2}$ | $\underline{54.5}_{\pm0.3}$ | $\underline{48.1}_{\pm0.2}$ | $\underline{50.4}_{\pm0.3}$ | $22.2_{\pm0.1}$ | 50.6 |
| **+ LogicTree** | $\mathbf{81.9}_{\pm0.2}$ | $\mathbf{52.3}_{\pm0.1}$ | $52.5_{\pm0.2}$ | $\mathbf{48.7}_{\pm0.2}$ | $\mathbf{51.4}_{\pm0.1}$ | $\mathbf{25.7}_{\pm0.2}$ | **52.1** |
| $\triangle$ **(Relative Gain)** | +4.4 (+5.7%) | +2.6 (+5.2%) | +1.3 (+2.5%) | +3.2 (+7.0%) | +1.5 (+3.0%) | +5.3 (+26.0%) | +3.1 (+6.3%) |
| **Llama-3.1-70B** | | | | | | | |
| vanilla | $91.3_{\pm0.2}$ | $67.1_{\pm0.2}$ | $59.6_{\pm0.4}$ | $75.3_{\pm0.2}$ | $82.2_{\pm0.1}$ | $28.3_{\pm0.1}$ | 67.3 |
| **+ PARARULE** | $92.5_{\pm0.3}$ | $68.2_{\pm0.2}$ | $61.1_{\pm0.3}$ | $75.6_{\pm0.1}$ | $82.4_{\pm0.2}$ | $26.1_{\pm0.2}$ | 67.7 |
| **+ LogicAsker** | $93.1_{\pm0.3}$ | $69.9_{\pm0.2}$ | $\underline{63.4}_{\pm0.2}$ | $75.2_{\pm0.1}$ | $82.8_{\pm0.2}$ | $\underline{30.8}_{\pm0.1}$ | 69.2 |
| **+ FLD$_{\times2}$** | $\underline{93.7}_{\pm0.2}$ | $\underline{69.9}_{\pm0.1}$ | $63.1_{\pm0.2}$ | $74.8_{\pm0.3}$ | $\underline{83.1}_{\pm0.3}$ | $26.1_{\pm0.1}$ | 68.5 |
| **+ LogicTree** | $\mathbf{94.4}_{\pm0.2}$ | $\mathbf{70.3}_{\pm0.1}$ | $\mathbf{65.2}_{\pm0.2}$ | $\mathbf{76.0}_{\pm0.0}$ | $\mathbf{83.7}_{\pm0.2}$ | $\mathbf{31.3}_{\pm0.1}$ | **70.2** |
| $\triangle$ **(Relative Gain)** | +3.1 (+3.4%) | +3.2 (+4.8%) | +5.6 (+9.4%) | +0.7 (+0.9%) | +1.5 (+1.8%) | +3.0 (+10.6%) | +2.9 (+4.3%) |

## 4.2 Main Results

To validate the significant improvement in logical reasoning abilities enabled by our synthesized data, we evaluate **LogicTree** on multiple downstream logical reasoning tasks. As highlighted in Table 1, **LogicTree** shows consistent and substantial performance improvements across all backbone models and most benchmarks against the baselines, demonstrating the effectiveness of instantiating complex logical reasoning trees. Specifically, LogicTree surpasses vanilla models on all benchmarks, achieving significant margins. On LogicBench and AGIEval-LR, it improves LLama-3.1-8B by 10.6% and 7.7%, respectively. Even on AGIEval-AR, a particularly challenging benchmark for LLMs, our approach still achieves a 6% improvement. Meanwhile, our method achieves average improvements of 9.4%, 3.1%, and 2.9% on Llama-3.1-8B, Mistral-7B-v0.3, and Llama-3.1-70B, respectively, underscoring its robustness across different model architectures and scales. Additional experimental results on more models can be found in Table 5 and 6.

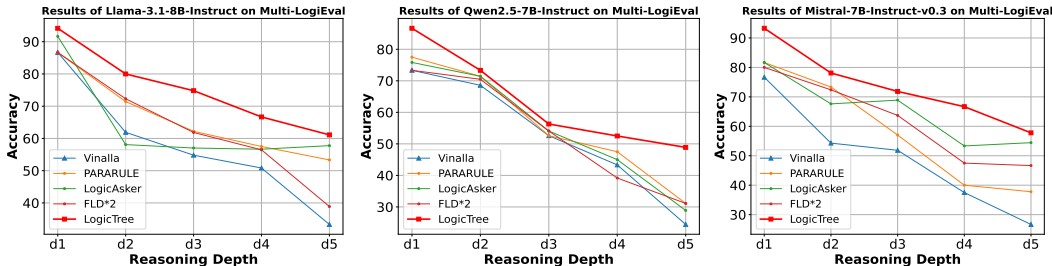

Figure 2: Experiment results for different methods on Multi-LogiEval, which is specifically designed to evaluate the model's multi-step logical reasoning abilities. The horizontal axis represents the number of reasoning steps required to answer the question, ranging from $d_1$ to $d_5$. The bold red line at the top represents our method, demonstrating superior performance improvements, especially as the number of reasoning steps increases.

Table 2: Experiment results using Llama3.1-8B across a broader range of task types, covering domains such as logic, math, code, NLI, and others.

| Model | Proofwriter | Mathqa | GPQA | Humaneval | Commencenseqa | MNLI | Avg. |
|---|---|---|---|---|---|---|---|
| **Llama-3.1-8B** | | | | | | | |
| vanilla | 57.3 | 42.1 | 30.8 | 68.4 | 73.9 | 68.1 | 56.8 |
| + PARARULE | 64.3 | 42.8 | 31.9 | 67.2 | 73.7 | 70.5 | 58.4 |
| + LogicAsker | 69.8 | 43.2 | 30.1 | 68.6 | 74.5 | 73.3 | 59.9 |
| + FLD$_{\times 2}$ | 74.1 | 43.8 | 31.1 | 72.3 | 74.1 | 75.3 | 61.8 |
| **+ LogicTree** | **76.9** | **47.3** | **32.8** | **74.6** | **75.9** | **76.8** | **64.1** |

It is worth noting that LogicTree demonstrates superior performance in all baselines, especially in multi-step complex reasoning tasks. While LogicAsker generates a large volume of atomic instruction data that provides some improvement in basic logical reasoning, it remains inferior to our approach. Furthermore, the datasets synthesized by these baselines yield only marginal gains on more challenging reasoning tasks, such as BBH-Logic and AGIEval-AR. In contrast, our method achieves performance gains of up to 13.9%, further demonstrating that **complex reasoning patterns and instantiated reasoning processes play a crucial role in enhancing the model's reasoning capabilities.**

### 4.3 Results of Multi-step Reasoning

To further evaluate the model's multi-step logical reasoning ability, we assess **LogicTree** and baselines on Multi-LogiEval, which categorizes questions by reasoning steps from one to five. As shown in Figure 2, LogicTree consistently outperforms all baselines across all categories of test data, with particularly strong improvements at higher reasoning depths (d3, d4, d5). These results indicate that LogicTree effectively enhances LLMs' ability to tackle complex multi-step reasoning tasks. In contrast, other methods exhibit a sharp decline in performance as reasoning depth increases, highlighting their limitations in scaling to more complex reasoning scenarios. Their struggle to maintain accuracy on multi-step reasoning tasks suggests that training with simple reasoning data alone is insufficient for significantly improving the model's ability to handle complex reasoning. LogicTree's superior performance on multi-step reasoning tasks underscores the potential benefits of its approach in improving reasoning abilities on multi-step tasks, making it a valuable tool for complex logical reasoning scenarios. Results related to the 70B model can be found in the Appendix B.4.

### 4.4 Results on Generalization Experiments Across Multiple Domains

To verify the generalization of LogicTree in enhancing model reasoning capabilities, we conducted additional experiments using an 8B model across a broader range of task types, covering domains such as logic, math, code, NLI, and others. As shown in the table, LogicTree enhances the model's reasoning abilities across multiple domains. The improvement is smaller in tasks requiring extensive

Table 3: Ablation results on four types of reasoning tasks, use Llama-3.1-8B as the backbone models. "Avg" means the average accuracy across four benchmarks for each methods. We bold the best results for each benchmark.

| Model | LogicBench | LogiQA2.0 | BBH | AGIEval | | Avg. |
| --- | --- | --- | --- | --- | --- | --- |
| | | | | LR | AR | |
| **Llama-3.1-8B** | | | | | | |
| vanilla | 80.0 | 42.4 | 39.3 | 48.4 | 20.7 | 46.2 |
| + LogicTree(num=3) | **90.6** | **53.5** | **53.2** | **56.1** | **26.5** | **55.9** |
| w/o inst_scenario | 86.3 | 53.5 | 46.7 | 54.5 | 22.6 | 52.7 |
| w/o inst_reasoning | 88.3 | 52.9 | 50.1 | 53.9 | 24.4 | 53.9 |
| w/o inst_diversity | | | | | | |
| num=1 | 83.1 | 51.2 | 49.6 | 54.1 | 23.9 | 52.4 |
| num=2 | 88.1 | 52.9 | 50.8 | 52.9 | 26.1 | 54.2 |
| num=4 | 92.6 | 52.7 | 53.2 | 56.1 | 25.7 | 55.2 |
| num=5 | 94.5 | 50.3 | 50.8 | 51.2 | 23.7 | 54.1 |

knowledge recall, such as commonsense and science, since LogicTree primarily strengthens generalization rather than knowledge retention, which is largely dependent on pretraining. However, in reasoning-intensive domains like logic, math, and code, the model exhibits significant gains, demonstrating that LogicTree effectively improves reasoning capabilities through diverse instantiated reasoning processes.

## 4.5 Ablation Study

In this section, we conduct ablation experiments to assess the contributions of key design components in LogicTree : (1) instantiating real-world reasoning scenarios (**Analysis 1**), (2) incorporating natural language reasoning processes under instantiated scenarios (**Analysis 2**), and (3) ensuring diversity in instantiated scenarios (**Analysis 3**). Specifically, "*w/o inst_scenario*" refers to training on symbolic logical expressions without instantiating the logical reasoning tree into real-world scenarios; "*w/o inst_reasoning*" denotes instantiating real-world scenarios while retaining a fully symbolic representation of the reasoning process; and "*inst_diversity*" means omitting control over the diversity of instantiated scenarios. In particular, we evaluate performance by instantiating one and two reasoning scenarios and processes per logical tree (whereas LogicTree employs three). As shown in Table3, removing any of these components results in a notable performance decline, highlighting their importance in enhancing logical reasoning capabilities.

**Analysis 1: Instantiating reasoning scenarios enables the model to learn the ability to apply logical reasoning in specific tasks, rather than simply memorizing language templates or the potential relationships between facts.** As shown in Table 3, while symbolic rules enhance reasoning, the improvement without instantiation is significantly less than with LogicTree. This underscores instantiation's importance: it teaches applying logic in complex tasks, not just memorizing rules.

**Analysis 2: The reasoning processes in natural language based on instantiated scenarios can further enhance the model's reasoning performance in challenging tasks.** As Table 3 demonstrates, instantiating symbolic logical problems into real-world scenarios improved the model's reasoning and accuracy compared to purely symbolic questions, supporting Analysis 1's findings. However, its performance remained inferior to LogicTree. This underscores that natural language reasoning based on instantiated scenarios is crucial for boosting logical ability in complex tasks. On hard problems like AGIEval-AR, this instantiation method yielded only modest gains 1.8%, highlighting the growing value of natural language reasoning on instantiated scenarios for intricate challenges.

**Analysis 3: Diverse reasoning scenarios able LLMs to develop more generalizable and transferable logical reasoning abilities.** We instantiate different numbers of reasoning scenarios for the same logical reasoning tree, ranging from 1 to 5. When num=1, the model's performance on some

benchmarks is even worse than when no instantiation is used ("w/o instantiation"). This suggests that using only a single reasoning scenario may cause the model to overfit to the relationships between the facts within that scenario, rather than learning genuine reasoning abilities. However, as num increases, the model's performance continues to improve and converges around 3 or 4. Therefore, incorporating more diverse reasoning scenarios proves beneficial for enabling the model to acquire genuinely generalized logical reasoning skills.

# 5 Related Works

## 5.1 Synthetic Logic Corpus for Training

With the increasing adoption of synthetic datasets [18, 9, 50], there has been a growing focus on generating high-quality logical reasoning datasets to enhance the reasoning capabilities of LLMs[53]. Early works[7, 39, 3] generated deductive processes by repeatedly applying deduction rules to a given set of facts and transformed these processes into natural language sentences using fixed templates. FLD*2 [30, 29] introduced systematic design principles for logical data synthesis and randomly combined various rules to construct multi-step reasoning datasets. **(author?)** [41] proposed a framework to evaluate the ability of LLMs to handle individual atomic reasoning rules and generate targeted samples for improvement. LogicPro [16] further advanced this line of research by using LLMs to convert algorithmic problems into logical reasoning tasks. However, These methods still have limitations in the complexity of logical trees, producing rules or facts with limited real-world applicability and simplistic language. Our research further extends the generation of logical reasoning trees, encompassing various complex reasoning patterns in first-order logic. Additionally, we use LLMs to instantiate these trees into realistic reasoning scenarios and generate reasoing process in natural language.

## 5.2 Evaluation of Logic Reasoning

Accurately evaluating the reasoning abilities of LLMs is both essential and challenging. Many studies have focused on assessing this fundamental capability of LLMs[38, 20]. For instance, LogiQA[19] and AR-LAST[55] evaluated models' logical reasoning abilities in real-world scenarios by collecting human examination questions. LogicBench [33] demonstrated that existing LLMs struggle to handle complex logical contexts, even when involving only a single reasoning pattern. **(author?)** [41] introduced LogicAsker, which utilizes a set of atomic reasoning skills to assess the ability of LLMs to learn logical rules. FOLIO[13] and Multi-LogiEval[35] further assessed the multi-step logical reasoning capabilities of LLMs. These studies [8, 4] highlight the significant challenges faced by LLMs in logical reasoning tasks, underscoring the critical need for high-quality training datasets to improve their reasoning performance.

# 6 Conclusion

In this paper, we propose **LogicTree**, a novel framework for synthesizing instantiated, complex logical reasoning datasets. Our method leverages first-order logic rules to generate complex multi-step logical reasoning trees, which are then instantiated using LLMs to produce natural language reasoning data with realistic scenarios. The resulting synthetic datasets feature intricate reasoning patterns, promoting the development of advanced reasoning abilities in LLMs to address complex tasks. Furthermore, by incorporating instantiated reasoning processes, our method enables LLMs to acquire generalizable reasoning skills, rather than simply memorizing implicit relationships between facts. Extensive experiments across multiple complex logical reasoning benchmarks demonstrate the effectiveness of LogicTree, with an average accuracy improvement of 9.4%. LogicTree consistently outperforms all baselines, particularly in multi-step complex reasoning tasks. Further analysis indicates that both instantiated reasoning processes and the diversity of instantiated scenarios contribute to enhancing the model's generalizable logical reasoning abilities.

## Acknowledgments

The authors would like to thank all the anonymous reviewers for their insightful comments and valuable suggestions. This work was supported by the National Key R&D Program of China under contract 2022ZD0119801, and the National Nature Science Foundations of China grants U23A20388, 62021001 and 624B1011.

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

**Impact Statement, Limitations and Future Work**

This paper follows existing research on logical reasoning data synthesis and constructs a multi-hop complex logical reasoning dataset for model training, which helps LLMs learn advanced and generalizable logical reasoning capabilities. Furthermore, our synthesized data can be used to precisely evaluate the model's multi-step logical reasoning ability, contributing to research on enhancing models' logical reasoning capabilities. Moreover, our work does not involve human or animal experiments, so the ethical impacts and expected societal implications are those that are well established.

This paper presents work whose goal is to advance the field of logical data synthesis. There are many potential societal consequences of our work, none of which we feel must be specifically highlighted here.

As for the limitations of LogicTree, while it significantly enhances the logical reasoning abilities of LLMs by synthesizing logical datasets, it cannot be integrated with other forms of reasoning. Exploring how to incorporate other reasoning paradigms, such as commonsense reasoning, into logical reasoning data to enhance the comprehensive reasoning abilities of LLMs will be the focus of our future work.

# A   More Related Works

## A.1   Prompt-Based Methods for Enhancing Logical Reasoning

Leveraging large language models (LLMs) for complex logical reasoning problems has been a key focus of recent research. Many efforts[12, 25] focus on designing more effective prompting strategies to help LLMs complete complex logical reasoning tasks. For example, Chain-of-Thought (CoT)[46] guides LLMs to reason step by step nd output the final answer based on the reasoning process. In order to better simulate rational logical thought processes, **(author?)** [54] proposed Diagram-of-Thought (DoT) that models iterative logical reasoning in LLMs as the construction of a directed acyclic graph (DAG). Other approaches involve prompting LLMs to first translate natural language into symbolic language, which is then used to facilitate logical reasoning. Symbolic Chain-of-Thought (SymbCoT) [49] converts input from natural language into a symbolic representation and subsequently constructs a step-by-step reasoning plan based on symbolic logic rules. Logic-of-Thought[23] employs LLMs to extract logical expressions and applies logical reasoning rules to derive extended logical expressions, which are then translated back into natural language to support subsequent reasoning.

Additionally, some research integrates LLMs with external tools, such as solvers, to enhance reasoning capabilities[26, 32]. These approaches first leverage LLMs to convert natural language input into a symbolic form (such as logic programming languages, first-order logic, constraint satisfaction problems, or Boolean satisfiability formulations) compatible with external solvers, which are then used to perform logical reasoning and yield the desired result.

Although prompt-based methods can effectively leverage the potential of large language models (LLMs), their performance remains fundamentally constrained by the models' inherent reasoning capabilities. This limitation is particularly evident in smaller models, which often produce hallucinations and logical errors during inference. On the other hand, approaches that incorporate external solvers primarily exploit the model's ability to translate natural language into symbolic representations, rather than enhancing its intrinsic reasoning ability. In contrast, our approach focuses on synthesizing a large amount of high-quality logical reasoning data from a training perspective, which helps enhance the inherent reasoning capabilities of LLMs.

## A.2   Reasoning Data Synthesis

An increasing number of studies have focused on synthesizing complex tasks and high-quality reasoning processes [43, 44, 22], which are utilized for supervised fine-tuning or preference alignment training[6, 17]. For example, the O1 series of studies leverages reinforcement learning and Monte Carlo Tree Search(MCTS) that enable LLMs to optimize reasoning processes autonomously. Benefiting from the extensive tree search space and the guidance of an excellent Process Reward Model, these methods can generate high-quality reasoning processes[52, 10, 48]. However, this approach incurs high computational costs and long search times, while generation efficiency is particularly

crucial in LLM-based settings[11]. In contrast, our method leverages rigorous symbolic logic trees to guide the LLM, offering lower costs and higher correctness.

## B   More Details about Experiments

### B.1   Information of Logical Reasoning Benchmarks

In this study, we selected six representative benchmarks to evaluate the reasoning capabilities of the models. These benchmarks encompass various aspects of logical reasoning, diverse contexts, and difficulty levels, making them suitable for a comprehensive evaluation. As shown in Table 4, we provide brief explanations of each benchmark.

Table 4: Information of Logical Reasoning Benchmarks used in our experiments

| Benchmark | Data Count | Explanation |
|---|---|---|
| LogicBench | 160 | LogicBench is a natural language dataset designed to evaluate the logical reasoning abilities of LLMs. In this work, we use the *MCQA* (Multiple Choice Question-Answering) subset of LogicBench(Eval), which focuses on tasks that apply a single inference rule. |
| LogiQA | 1527 | LogiQA 2.0 is an enhanced dataset for evaluating logical reasoning in natural language understanding tasks. We use the main part of LogiQA, which includes multiple-choice reading comprehension questions. |
| FOLIO | 203 | FOLIO is an expert-crafted dataset for evaluating natural language reasoning with first-order logic. It includes logically complex examples presented in natural language and their formal FOL representations. For our experiment, we use all validation data, leveraging its dual-format structure to precisely assess models' ability to interpret and reason with formal logical constructs. |
| BBH | 1187 | BIG-Bench Hard (BBH) is a subset of 23 challenging tasks from the BIG-Bench benchmark, focusing on advanced reasoning. For our experiment, we use three tasks: *Causal Judgment*, evaluating causal reasoning in stories; *Formal Fallacies Syllogisms Negation*, testing logical consistency in argument schemes; and *Logical Deduction*, assessing sequence deduction skills. |
| AGIEval | 740 | AGIEval is a benchmark assessing foundation models' abilities through tasks from high-standard exams. For our experiment, we use the *LSAT* (Law School Admission Test) tasks, which evaluate logical reasoning, reading comprehension, and analytical reasoning. These tasks challenge models to analyze complex information and draw accurate conclusions, providing a valuable assessment of their capabilities in legal reasoning and analysis. |
| Multi-LogiEval | 525 | Multi-LogiEval is a dataset designed to evaluate LLMs' multi-step reasoning abilities across various logic types. This focus allows for a precise evaluation of LLMs' capabilities in handling logical constructs while maintaining manageable complexity. |

### B.2   Experiment Setups

This section details our experimental design, covering model selection and the specific training configurations. To rigorously validate the effectiveness of LogicTree, we conduct extensive experiments using a diverse set of prominent open-source models spanning various families and scales.

We utilize models from the Llama-3.1, Mistral-v0.3, Qwen2.5, and Deepseek-R1-Distill families, with parameter scales ranging from 1.5B to 70B. We employ two distinct fine-tuning strategies: full fine-tuning is applied to smaller-scale models (under 8B), while the larger 70B model utilizes LoRA fine-tuning. Llama-3.1-8B, Mistral-7B-v0.3 and Qwen2.5-7B are both trained with a learning rate of $1e-6$. Qwen2.5-1.5B and Qwen2.5-3B are trained with a learning rate of $3e-6$. Llama-3.1-70B, due to its LoRA fine-tuning method, is trained with a higher learning rate of $2e-5$. The training utilizes a maximum context length of 4096 tokens, a global batch size of 128, and is conducted for 3 epochs. DeepSpeed with gradient checkpointing and BF16 precision is used for efficient memory usage. The learning rate scheduler follows a cosine schedule, and the warmup ratio is set to 0.03.

### B.3    More results of Different Models

To comprehensively validate the effectiveness of LogicTree, We use advanced reasoning models such as DeepSeek-R1-Distill-Llama-8B and DeepSeek-R1-Distill-Qwen-7B. As shown in the table 5, LogicTree consistently improves the performance of DeepSeek-R1-Distill-Llama-8B and DeepSeek-R1-Distill-Qwen-7B across multiple benchmarks, with average gains of 6.5 % and 4.9 %, respectively. Notably, on logic-intensive benchmarks such as the ML series, the improvements reach up to 13.3 % and 10.9 %. These results further confirm the effectiveness of LogicTree in enhancing the logical reasoning capabilities of LLMs.

Table 5: Results of DeepSeek-R1-Distill series models across multiple logical reasoning benchmarks.

| Model | Dataset | | | | | | | | | | Avg |
| --- | --- | --- | --- | --- | --- | --- | --- | --- | --- | --- | --- |
| | LogicB. | LogIQA | FOLIO | AGIEval-LR | AGIEval-AR | ML-D1 | ML-D2 | ML-D3 | ML-D4 | ML-D5 | |
| *DeepSeek-R1-Distill-Qwen-7B* | | | | | | | | | | | |
| Vanilla | 86.8 | 52.4 | 61.2 | 56.8 | 32.6 | 70.0 | 57.0 | 60.0 | 55.0 | 44.4 | 57.6 |
| +LogicTree | 90.6 | 53.4 | 63.4 | 62.4 | 34.4 | 80.9 | 67.6 | 62.5 | 56.6 | 53.3 | 62.5 |
| Δ | +3.8 | +1.0 | +2.2 | +5.6 | +1.8 | +10.9 | +10.6 | +2.5 | +1.6 | +9.9 | +4.9 |
| *DeepSeek-R1-Distill-Llama-8B* | | | | | | | | | | | |
| Vanilla | 83.1 | 53.4 | 53.1 | 53.3 | 30.4 | 78.0 | 67.1 | 60.0 | 49.2 | 46.6 | 57.4 |
| +LogicTree | 91.2 | 54.9 | 57.8 | 57.3 | 33.9 | 86.3 | 73.3 | 66.0 | 62.5 | 56.6 | 63.9 |
| Δ | +8.1 | +1.5 | +4.7 | +4.0 | +3.5 | +7.3 | +6.2 | +6.0 | +13.3 | +10.0 | +6.5 |

In addition,we conduct additional experiments using Qwen2.5-1.5B and Qwen2.5-3B to further evaluate the effectiveness of LogicTree on smaller language models. As shown in the Table 6, LogicTree effectively enhances the logical reasoning capabilities of smaller models such as Qwen2.5-1.5B and Qwen2.5-3B, achieving consistent and substantial improvements across multiple benchmarks. Specifically, Qwen2.5-1.5B achieves an average accuracy gain of 7.7 %, and Qwen2.5-3B improves by 9.3 %, demonstrating that our synthesized data remains effective even for smaller-scale models.

Table 6: Results of small models across multiple logical reasoning benchmarks.

| Model | Dataset | | | | | | | | | | Avg |
| --- | --- | --- | --- | --- | --- | --- | --- | --- | --- | --- | --- |
| | LogicBench | LogIQA2.0 | FOLIO | AGIEval-LR | AGIEval-AR | ML-D1 | ML-D2 | ML-D3 | ML-D4 | ML-D5 | |
| *Qwen2.5-1.5B* | | | | | | | | | | | |
| Vanilla | 65.0 | 43.9 | 48.5 | 39.5 | 17.4 | 65.6 | 53.3 | 48.8 | 43.3 | 31.1 | 45.8 |
| +LogicTree | 74.4 | 48.4 | 53.4 | 42.4 | 21.3 | 80.9 | 56.2 | 54.8 | 54.2 | 44.4 | 53.5 |
| *Qwen2.5-3B* | | | | | | | | | | | |
| Vanilla | 74.4 | 53.9 | 51.0 | 53.5 | 20.9 | 74.5 | 55.2 | 46.5 | 45.0 | 36.6 | 51.2 |
| +LogicTree | 83.9 | 56.2 | 61.2 | 55.7 | 24.4 | 86.3 | 67.6 | 59.3 | 58.3 | 50.0 | 60.5 |

### B.4    More Results on Multi-LogiEval

We have further conducted additional experiments on multi-step reasoning(sec4.3) using LLaMA3.1-70B to validate the robustness of our method. Although the 70B model shows some performance improvement over the 8B model at various reasoning depths, it still faces a decline in accuracy as the number of reasoning steps increases. When the reasoning depth reaches 5, the model's accuracy is only 44%. Our method significantly enhances the model's logical reasoning accuracy across different reasoning depths, demonstrating its superiority. As shown in the table 7, the LLaMA3.1-70B model achieved notable performance, even surpassing the vanilla model and all baseline methods, which highlights the scalability of our approach.

Table 7: Results of Llama-3.1-70B on Multi-LogiEval

| Model | Method | d1 | d2 | d3 | d4 | d5 |
|---|---|---|---|---|---|---|
| Llama-3.1-8B | vanilla | 86.67 | 61.90 | 54.81 | 50.83 | 33.33 |
| | PARARULE | 86.67 | 71.43 | 62.22 | 57.50 | 53.33 |
| | logicAsker | 91.66 | 58.09 | 57.03 | 56.66 | 57.77 |
| | FLD$_{\times 2}$ | 86.67 | 72.31 | 61.85 | 56.47 | 38.89 |
| | **LogicTree** | **94.17** | **80.00** | **74.81** | **66.67** | **61.11** |
| Mistral-7B-v0.3 | vanilla | 76.67 | 54.29 | 51.85 | 37.50 | 26.67 |
| | PARARULE | 81.67 | 73.33 | 57.04 | 40.00 | 37.78 |
| | logicAsker | 81.67 | 67.62 | 68.89 | 53.33 | 54.44 |
| | FLD$_{\times 2}$ | 80.00 | 72.38 | 63.70 | 47.50 | 46.67 |
| | **LogicTree** | **93.30** | **78.10** | **71.85** | **66.67** | **57.78** |
| Qwen2.5-7B | vanilla | 73.33 | 68.57 | 52.59 | 43.33 | 24.44 |
| | PARARULE | 77.50 | 71.43 | 52.59 | 47.50 | 31.11 |
| | logicAsker | 75.83 | 71.43 | 54.07 | 45.00 | 28.89 |
| | FLD$_{\times 2}$ | 73.33 | 70.48 | 54.07 | 39.17 | 31.11 |
| | **LogicTree** | **86.67** | **73.33** | **56.30** | **52.50** | **48.89** |
| Llama-3.1-70B | vanilla | 90.8 | 64.7 | 58.2 | 61.7 | 44.4 |
| | PARARULE | 93.3 | 69.5 | 59.3 | 63.3 | 56.7 |
| | LogicAsker | 93.3 | 69.5 | 60.0 | 65.8 | 46.7 |
| | FLD$_{\times 2}$ | 90.8 | 66.7 | 63.9 | 61.7 | 46.7 |
| | **LogicTree** | **96.7** | **73.3** | **76.3** | **74.2** | **60.0** |

## B.5 Results on Quantitative Evaluation of Synthetic Data

Table 8: Evaluation of logical consistency of synthetic data across multiple models

| Model | Consistency |
|---|---|
| GPT-4o | 97.6% |
| Deepseek-R1 | 96.8% |

**Logical Consistency**   To minimize instantiation errors as much as possible, we designed a two-stage prompting strategy to guide the LLM in instantiating logical reasoning trees. Through this process, the LLM does not need to perform reasoning on its own; it only needs to perform symbolic translation. Additionally, we employed a systematic verification method to filter out errors and filtered out 8.73% of erroneous data, thereby improving the overall data quality. We also conducted additional experiments to evaluate the logical consistency of the filtered instantiated data. Specifically, we prompted multiple LLMs to independently assess whether the logical expressions and their corresponding natural language statements were logically consistent. As shown in Table 8, our data maintained a high level of logical consistency across evaluations by several state-of-the-art models. The prompt used for evaluation with LLMs is as follows:

```
You are an expert in logical reasoning and formal logic.  Based on the
correspondence between entities and logical symbols, your task is to
evaluate the consistency between the given natural language statements
and corresponding logical expressions.  Please assess whether the logical
expressions accurately translate the logical semantics of the natural
language statements.
```

**Realism & Contextual Richness**   We conducted additional validation experiments to demonstrate the superiority of our synthesized data in terms of realism and contextual richness. Specifically, we employed multiple LLMs to evaluate the synthesized data and compared the results with other methods. The evaluation metrics are defined as follows:

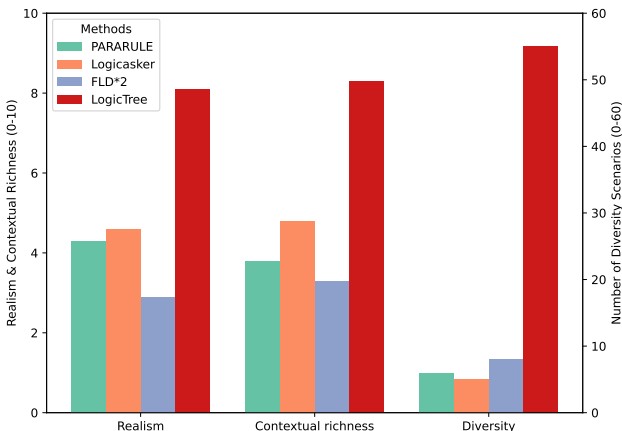

Figure 3: Comparison of Different Methods Across Realism, Contextual Richness, and Diversity. The red bars represent LogicTree, which significantly outperforms other baseline methods.

1. Realism: Assess whether the data presents a realistic, plausible, and logically consistent real-world scenario.

2. Contextual richness: Assess whether the data includes diverse and detailed elements that enrich the scenario.

Notably, due to the rich world knowledge and strong language capabilities of LLMs, LogicTree achieved the highest score for realism and contextual richness in Figure 3. LogicTreeground synthetic reasoning data in concrete and context-rich real-world scenarios that incorporate semantically relevant entities or events, capable of reflecting the complexity and diversity of real-world reasoning tasks. The prompts used for evaluation with LLMs are as follows:

```
You are an expert in data evaluation, contextual analysis, and scenario
validation.  Your task is to evaluate the contextual richness of the
generated data, assigning a single score out of 10 to reflect the overall
quality.
The evaluation should consider the following criteria:
1.  Contextual Depth:  Assess whether the data includes diverse and
detailed elements that enrich the scenario.
2.  Variety of Information:  Determine whether the generated data presents
a broad range of relevant contextual elements.
Scoring Guidelines:
• 10:  Very contextually rich with a wide variety of relevant information.
• 9:  Generally rich in context, with sufficient variety and depth, though
slightly lacking in some aspects.  • 8:  Moderately rich with noticeable
diversity, but certain areas could be expanded to enhance contextual
richness.  • 6:  Provides basic context with some variety.  • 5:  Limited
contextual richness, with minimal diversity or detailed elements.  • 3:
Poorly developed with very few diverse contextual elements, resulting in a
shallow scenario.  • 1:  Lacks any meaningful contextual richness, offering
almost no variety or depth.
```

```
You are an expert in You are an expert in data evaluation, logical
reasoning, and real-world scenario validation.  Your task is to evaluate
the realism of the generated data, assigning a single score out of 10 to
reflect the overall quality.  The evaluation should consider the following
criteria:
1.  Realism:  Assess whether the data presents a realistic and plausible
real-world scenario.
2.  Logical Consistency:  Determine whether the generated data maintains
internal logical coherence.
Scoring Guidelines:
• 10:  Highly realistic and plausible.  • 9:  Mostly realistic and
consistent with minor deviations that do not affect overall plausibility.
• 8:  Generally realistic and coherent with slight inconsistencies
or plausible.  • 7:  Somewhat realistic but with inconsistencies that
slightly affect coherence or plausibility.  • 6:  Reasonably realistic
but with noticeable flaws that slightly impact plausibility and background
consistency..  • 5:  Lacks realism and consistency, with major flaws in
logical coherence and background alignment.  • 3:  Highly unrealistic and
implausible, with multiple errors and misalignments.  • 1:  Completely
unrealistic and logically incoherent.
```

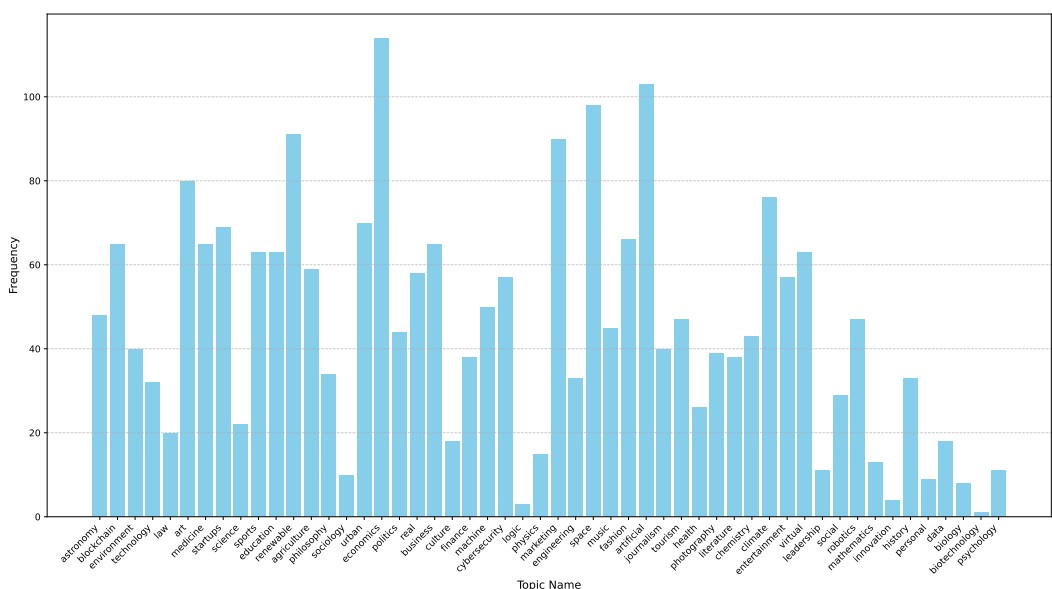

Figure 4: Statistical analysis of the diversity of the synthesized scenarios. We summarized over 50 common themes and prompted the LLM to instantiate multiple real-world scenarios for each logical reasoning tree.

**Diversity Scenarios**  Our goal in introducing instantiation diversity is to enable the model to truly learn logical reasoning rules rather than simply memorizing implicit relationships between specific contents.  To achieve this, we summarized over 50 common themes and prompted the LLM to instantiate multiple real-world scenarios for each logical reasoning tree. We also provided a statistical analysis(Figure 4) of the diversity of the synthesized scenarios to demonstrate the effectiveness of this approach.

## C  More Details about Logical Reasoning Tree Generation

### C.1  Logical Reasoning Tree Generation Process

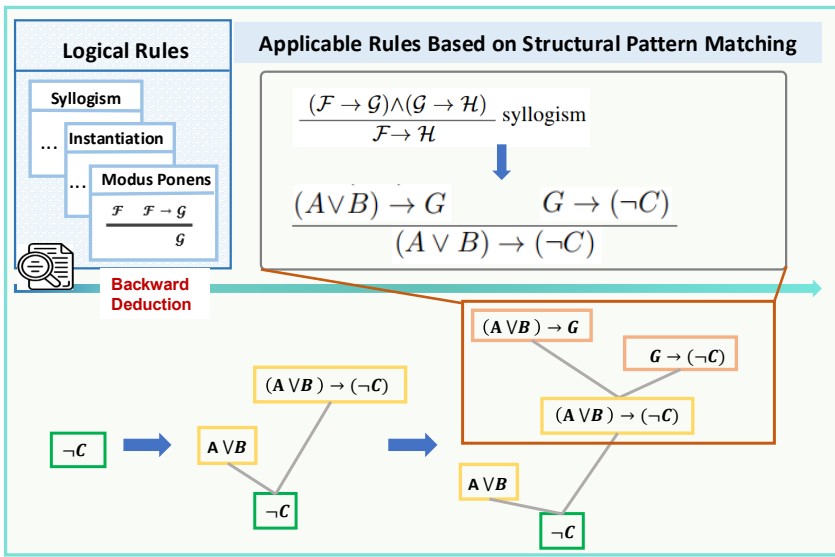

Figure 5: More Details about Logical Reasoning Tree Generation

To clarify our computational cost, we analyzed the number of LLM output tokens consumed when generating each data instance. Specifically, assuming that each reasoning step consumes $T$ output tokens, the computational cost of generating a data instance with reasoning steps $m$ is as follows for each method. We define **Instantiation Cost** and **Verification Cost**.

1. For the **Instantiation Cost**:1) The first call to LLM populates the abstract logical symbols with concrete natural language entities. We denote the number of output tokens consumed in this step as $c$. 2) The second call translates the sequence of reasoning steps into natural language. We assume that translating each individual reasoning step requires an average of output tokens$T$. The computational cost of generating a data instance with reasoning steps $m$ is:
$$Cost_{inst} = m \cdot T + c$$

2. For the **Verification Cost**:We assume that the complexity of verifying a single step is comparable to that of translating it during instantiation. Consequently, the computational cost for verifying a data instance with $m$ reasoning steps is:
$$Cost_{ver} = m \cdot T$$

.

As shown in the table 9 (the data in the parentheses), the ratio of synthesized tokens to consumed tokens for our method is $\frac{mT}{2mT+c}$ (close to 1/2, in our data $T = 79 \pm 14$, $c = 114 \pm 44$), which is significantly lower than the cost of other methods. Our framework's guarantee of correctness eliminates the need for multiple LLM calls. Consequently, its generation cost is entirely acceptable in comparison with other approaches.

### C.2  Statistical Information of Key Properties for logical synthetic data

In order to construct logical reasoning tree, previous methods typically concatenate rules iteratively from a predefined set and randomly replace atomic formulas with complex ones, resulting in an uncontrollable generation process. Moreover, rules are only concatenated when the conclusion of one

Table 9: A comparison of the generation costs for different LLM-based data synthesis methods.

| Method | V-STaR | ALPHALLM | LogicTree |
|---|---|---|---|
| **Number of LLM invocations** | $k$ | $mbk$ | 3 |
| **Number of consumed LLM output tokens** | $mkt$ | $mbk(T_{\text{rollout}}) + mT$ | $2mT + c$ |
| **Number of synthesized tokens** | $mT$ | $mT$ | $mT$ |
| **Explanation** | Where $k$ is the number of candidate reasoning trajectories. | Where $b$ is the number of candidate reasoning nodes per step, $k$ is the number of rollout simulations for each node, and $T_{\text{rollout}}$ is the additional tokens consumed during the rollout process. | where $c$ denotes the tokens consumed during the first invocation to generate the instantiated entities. |

matches the premise of another, which restricts the diversity and complexity of reasoning patterns (e.g., preventing the integration of propositional and first-order logic rules).

In contrast, we designed a backward deduction method based on structural pattern matching of formulas, which does not require the formula to be identical to the conclusion of the logic rule; instead, it only compares whether the structural patterns of their are matched. In this way, we generate multi-step logical reasoning trees that incorporate diverse and complex reasoning patterns, encompassing both propositional logic and first-order logic. In Table 10, we provide additional details about our synthetic data and compare it with other methods. And in table 11, we present supplementary statistical information to further characterize our dataset.

Table 10: The comparison of synthetic logic corpora, which focuses on several key characteristics: the number of logical rules, the reasoning depth, the symbolic-to-natural language translation method, and the instantiation approach.

| | Logic Rules | Reasoning Steps | Translation | Instantiation |
|---|---|---|---|---|
| RuleTaker | 2 | 1-5 | Template | Random Entities |
| PARARULE | 2 | 1-5 | Template | Random Entities |
| FLD | 13 | 1-8 | Template | WorldNet |
| FLD$_{\times 2}$ | $\approx 50$ | 1-8 | Template | WorldNet |
| **LogicTree** | **190** | **1-15** | **LLM-based** | **Realistic Scenario** |

Table 11: Statistical information for logic trees with 5 to 8 reasoning steps, including (1) the average maximum depth of the trees, (2) the average number of nodes per tree, (3) the average number of distinct rules applied, as well as (4) the proportion of first-order logic rules and propositional logic rules used in each tree.

| Step | Number of Nodes | Maximum Depth | Number of Distinct rules | Proportion of Fol | Proportion of Prop |
|---|---|---|---|---|---|
| 5 | $11.46 \pm 1.19$ | $3.68 \pm 1.42$ | $4.14 \pm 0.52$ | 31.42 | 68.58 |
| 6 | $13.46 \pm 1.01$ | $5.04 \pm 0.32$ | $4.75 \pm 0.81$ | 30.45 | 69.55 |
| 7 | $15.78 \pm 1.81$ | $5.61 \pm 0.56$ | $5.29 \pm 0.72$ | 29.94 | 70.06 |
| 8 | $18.30 \pm 2.01$ | $6.04 \pm 0.76$ | $5.67 \pm 0.97$ | 28.82 | 71.18 |

### C.3 Logical Rules used in LogicTree

This section presents the logical rules employed in the LogicTree generation process. These rules are fundamental to automated reasoning and inference generation, serving as the foundation for constructing structured logical trees. Drawn from both First-order Logic and Propositional Logic, the applied rules encompass a variety of logical inference patterns, including Modus Ponens, Hypothetical Syllogism, Disjunctive Syllogism, and Universal Instantiation. Table 12 presents the symbolic form and natural language explanation for a selection of common logical rules.

Table 12: Some Logical Rules Used in LogicTree Generation Process and Their Explanations

| Rule Name | Rule Symbol | Explanation |
|---|---|---|
| **First-order Logic** | | |
| MP | $(\forall x(p(x) \to q(x)) \wedge p(a)) \vdash q(a)$ | Modus Ponens: Universal elimination combined with conjunction provides the result $q(a)$. |
| MT | $(\forall x(p(x) \to q(x)) \wedge \neg q(a)) \vdash \neg p(a)$ | Modus Tollens: From universal elimination and negation, derives $\neg p(a)$. |
| HS | $(\forall x((p(x) \to q(x)) \wedge (q(x) \to r(x)))) \vdash (p(a) \to r(a))$ | Hypothetical Syllogism: Nested implications for quantified variables result in the conditional $p(a) \to r(a)$. |
| DS | $(\forall x(p(x) \vee q(x)) \wedge \neg p(a)) \vdash q(a)$ | Disjunctive Syllogism: Disjunction with universal quantification simplifies to $q(a)$ if $\neg p(a)$ is given. |
| UI | $\forall x(p(x)) \vdash p(a)$ | Universal Instantiation: From the universal quantifier $\forall x(p(x))$, deduce the specific instance $p(a)$. |
| **Propositional Logic** | | |
| MP | $((p \to q) \wedge p) \vdash q$ | Modus Ponens: If $p \to q$ and $p$, then $q$. |
| DS | $((p \vee q) \wedge \neg p) \vdash q$ | Disjunctive Syllogism: If $p \vee q$ and $\neg p$, then $q$. |
| MT | $((p \to q) \wedge \neg q) \vdash \neg p$ | Modus Tollens: If $p \to q$ and $\neg q$, then $\neg p$. |
| HS | $((p \to q) \wedge (q \to r)) \vdash (p \to r)$ | Hypothetical Syllogism: If $p \to q$ and $q \to r$, then $p \to r$. |
| MI | $(p \to q) \dashv\vdash (\neg p \vee q)$ | Material Implication: Expresses conditional as disjunction. |
| DMT | $\neg(p \wedge q) \dashv\vdash \neg p \vee \neg q$ | De Morgan's Theorem: Simplifies negation of conjunctions. |
| CD | $((p \to q) \wedge (r \to s) \wedge (p \vee r)) \vdash (q \vee s)$ | Constructive Dilemma: Combines conditional and disjunctive reasoning. |
| DD | $((p \to q) \wedge (r \to s) \wedge (\neg q \vee \neg s)) \vdash (\neg p \vee \neg r)$ | Destructive Dilemma: If one or more results fail, one or more premises fail. |
| BD | $((p \to q) \wedge (r \to s) \wedge (p \vee \neg s)) \vdash (q \vee \neg r)$ | Bipolar Dilemma: Variation of dilemma reasoning combining conditions. |

# D   Details about Prompts for Scenario Instantiation

This section discusses the prompts used during the reasoning scenario instantiation step, detailing their structure and role in guiding the model to produce relevant and coherent outputs for logic trees.

## D.1   Logical Tree Instantiation

As shown in Figure 6, our logical tree instantiation prompt primarily consists of five components: 1. Role-play: Setting the context for the model to assume a relevant role. 2. Instantiation instruction: Guiding the model to instantiate logical symbols using appropriate real-world scenarios. 3. Logical expression translation instruction: Assisting the model in understanding key logical relationships by providing in-context learning examples for translation. 4. Diversity control: Selecting different instantiation scenarios to ensure diversity. 5. Specific instantiation in-context learning examples: Offering concrete examples to further guide the instantiation process.

> You are a highly skilled logic analyst with expertise in understanding and interpreting complex logical relationships.
>
> Please deeply understand the logical rules between the propositions. Note the meanings of the logical symbols:
>    - '~' (NOT): Negation, indicates the proposition is not true.
>    - '>' (IMPLIES):Implies, indicates that there is a inferential relationship between the two propositions.
>    -'|' (OR): Logical disjunction, indicates that at least one of the propositions on either side is true.
>    - '&' (AND): Logical conjunction, indicates that both propositions on either side are true.
>    - '∀x' (FOR ALL): Universal quantifier, indicates that a statement is true for all values of x.
> While satisfying the logical rules, replace the symbolic propositions with real life complex events from the most relevant field. Please think step by step to ensure that the replaced events can naturally satisfy the logic rules.
>
> Use the corresponding events to explain each logic rule expression in natural language. Please follow the priority order in the logical expression, think step by step, and ensure the translation is accurate and clear. For example:
>    i: P1>(~((~P0)&Q))----If P1 is true, either P0 is true, or Q is false, or both.
>    ii: (~(Q|R0))>P2----If both Q and R0 are false, then P2 must be true.
>    iii:(~P2)|(P0|Q)----Either P2 is false, or at least one of P0 or Q is true.
>    iv: ((P1&P2)>((~Q)>P0))----If both P1 and P2 are true, then [if Q is false, P0 must also be true.].
>    v: (∀x(P4(x)))----For all x, P4(x) is true.
>
> Generate 3 examples from multiple domains [{domain}], We hope you can leverage relevant entities in this field to construct the aforementioned complex events.
>
> -----------example1------------{example1}
> -----------example2------------{example2}
> -----------example3------------{example3}

Figure 6: Prompt for Logical Tree Instantiation

## D.2 Reasoning Process Generation

As shown in Figure 7, our reasoning process generation prompt primarily consists of five components: 1. Role-play: Setting the context for the model to assume a relevant role. 2. Understanding of logical relations: It requires recognizing logical operators, understanding how propositions are structured, and determining how they interact based on formal logic rules. 3. Step-by-Step Reasoning Process Generation: In this stage, a structured sequence of logical steps is generated, following formal inference rules. Each step should be clearly justified to show how the conclusion is logically derived from the premises. 4. Conclusion and Answer Explanation: This part provides the final outcome derived from the reasoning process and offers a concise explanation of why the conclusion is valid. 5. Specific reasoning process generation learning examples: Offering concrete examples to further guide the instantiation process.

You are a highly skilled logic analyst with expertise in understanding and interpreting complex logical relationships.

The following is a logical reasoning context and some conclusions. For each conclusion,the truth value can be Yes, No or Uncertain. And We have extracted the main event entities and logic expressions from the context and conclusions. Note the meanings of the logical symbols:
   - '~' (NOT): Negation, indicates the proposition is not true.
   - '>' (IMPLIES):Implies, indicates that there is a inferential relationship between the two propositions.
   - '|' (OR): Logical disjunction, indicates that at least one of the propositions on either side is true.
   - '&' (AND): Logical conjunction, indicates that both propositions on either side are true.
   - '∀x' (FOR ALL): Universal quantifier, indicates that a statement is true for all values of x.

Please refer to the following reasoning steps and the extracted entities to provide a detailed reasoning explanation in natural language. For each step of reasoning, we provide the given premises and the conclusions that can be deduced. Please carefully explain each step in detail.

for each conclusion, refer to the reasoning process generated above and provide an explanation that aligns with the corresponding answer.

-----------example1-----------{example1}
-----------example2-----------{example2}
-----------example3-----------{example3}

Figure 7: Prompt for Reasoning Process Generation

# E  Examples of Synthetic Logical Reasoning data

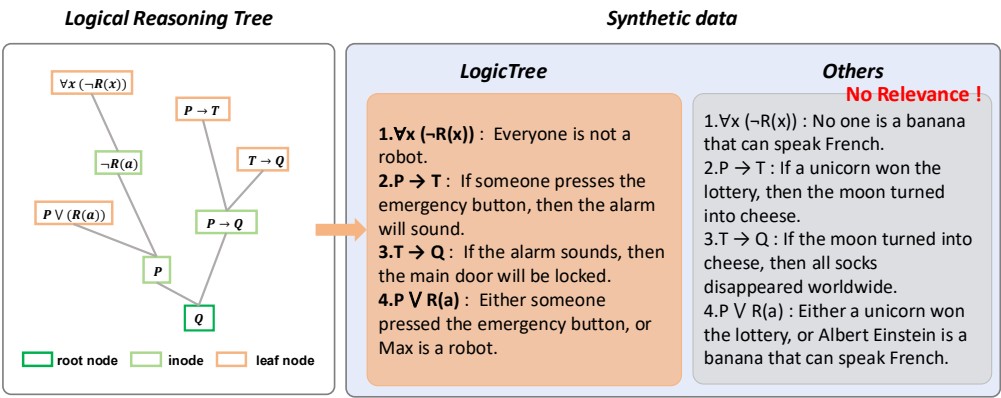

Figure 8: The logical reasoning trees and the corresponding synthetic data are presented, with the left side generated by LogicTree and the right side produced using a template-based random substitution method. It can be observed that the data synthesized by LogicTree maintains a coherent contextual semantic relationship, whereas the data generated by other methods is either unrelated or even contradictory.

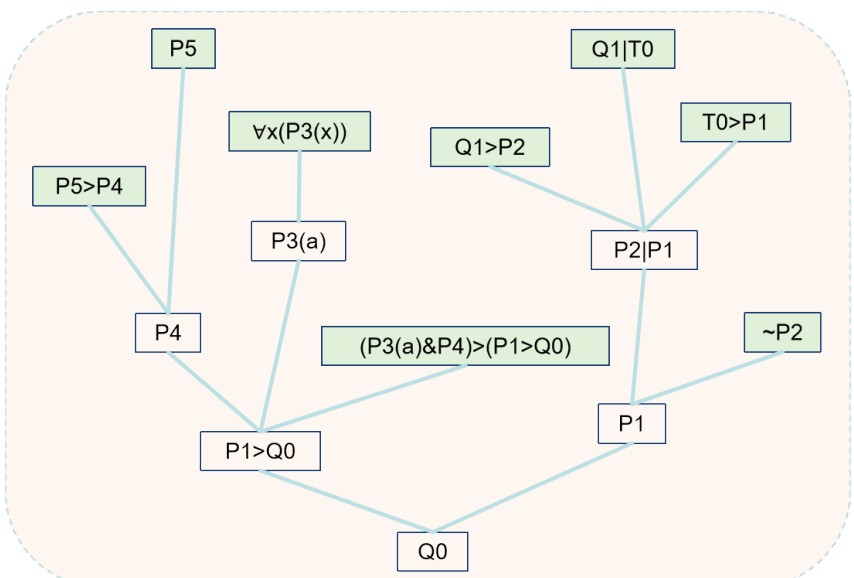

Figure 9: An example of a more complex logical reasoning tree, where green blocks implying the leaves to generate scenarioes

**Entities:**
- **P1**: Emergency shelters were opened in time.
- **P2**: Casualties occurred.
- **P3(x)**: x is located in a high-risk seismic zone.
- **P4**: Local authorities activated the emergency response protocol.
- **P5**: A magnitude 6.0 or greater earthquake was detected.
- **Q0**: People were safely evacuated.
- **Q1**: Warning sirens were activated.
- **T0**: The national emergency broadcast was triggered.
- **a**: City A

**Rules:**
rule1: $((P3(a) \land P4) \rightarrow (P1 \rightarrow Q0))$ If City A is in a high-risk seismic zone and local authorities activated the emergency response protocol, then if emergency shelters were opened in time, people were safely evacuated.
rule2: $\forall x(P3(x))$ All regions are located in high-risk seismic zones.
rule3: $(P5 \rightarrow P4)$ If a magnitude 6.0 or greater earthquake was detected, then the emergency response protocol was activated.
rule5: $(Q1 \rightarrow P2)$ If warning sirens were activated, then casualties occurred.
rule6: $(T0 \rightarrow P1)$ If the national emergency broadcast was triggered, then emergency shelters were opened in time.
rule7: $(Q1 \lor T0)$ Either warning sirens were activated, or the national emergency broadcast was triggered.
rule8: $\neg P2$ No casualties occurred.

**Reasoning process:**

**Step1:** From rule5 $(Q1 \rightarrow P2)$ [if warning sirens were activated, then casualties occurred], rule7 $(Q1 \lor T0)$ [either warning sirens were activated or the national emergency broadcast was triggered], and rule6 $(T0 \rightarrow P1)$ [if the national emergency broadcast was triggered, then emergency shelters were opened in time], we can deduce that $(P2 \lor P1)$ [either casualties occurred or emergency shelters were opened in time].
**Step2:** According to rule2 $(\forall x(P3(x)))$ [all cities are located in high-risk seismic zones], we can conclude that $(P3(a))$ [City A is located in a high-risk seismic zone].
**Step3:** From rule3 $(P5 \rightarrow P4)$ [if a magnitude 6.0 or greater earthquake was detected, then the emergency response protocol was activated] and assuming $(P5)$ [a magnitude 6.0+ earthquake was detected], we can conclude that $(P4)$ [the emergency response protocol was activated].
**Step4:** According to rule1 $((P3(a) \land P4) \rightarrow (P1 \rightarrow Q0))$ [if City A is in a high-risk seismic zone and the emergency response protocol was activated, then if emergency shelters were opened in time, people were safely evacuated], and from step2 $(P3(a))$ and step3 $(P4)$, we can deduce that $(P1 \rightarrow Q0)$ [if emergency shelters were opened in time, people were safely evacuated].
**Step5:** From step1 $(P2 \lor P1)$ [either casualties occurred or emergency shelters were opened in time] and rule8 $(\neg P2)$ [no casualties occurred], we can conclude that $(P1)$ [emergency shelters were opened in time].
**Step6:** From step4 $(P1 \rightarrow Q0)$ [if emergency shelters were opened in time, people were safely evacuated] and step5 $(P1)$ [emergency shelters were opened in time], we can conclude that $(Q0)$ [people were safely evacuated].

Figure 10: The Generated Entities, Rules and Reasoning Process of an Example

**conclusion1:** people were safely evacuated.
**answer:** Yes
**explanation:** From step6, we deduced that people were safely evacuated (Q0). Therefore, the conclusion that people were safely evacuated is true.

**conclusion2:** emergency shelters were not opened in time
**answer:** No
**explanation:** From step5, we deduced that emergency shelters were opened in time (P1). Therefore, the conclusion that emergency shelters were not opened in time is false.

**conclusion3:** either casualties occurred or emergency shelters were opened in time
**answer:** No
**explanation:** From step1, we deduced that either casualties occurred or emergency shelters were opened in time (P2 ∨ P1). From step5, we know emergency shelters were opened in time (P1), making the statement "It is not the case that either casualties occurred or emergency shelters were opened in time" false.

**conclusion4:** it is not the case that if emergency shelters were opened in time, people were safely evacuated
**answer:** No
**explanation:** From step4, we deduced that if emergency shelters were opened in time, people were safely evacuated (P1 → Q0). Therefore, the conclusion that it is not the case that if emergency shelters were opened in time, people were safely evacuated is false.

Figure 11: The Generated Answers of an Example

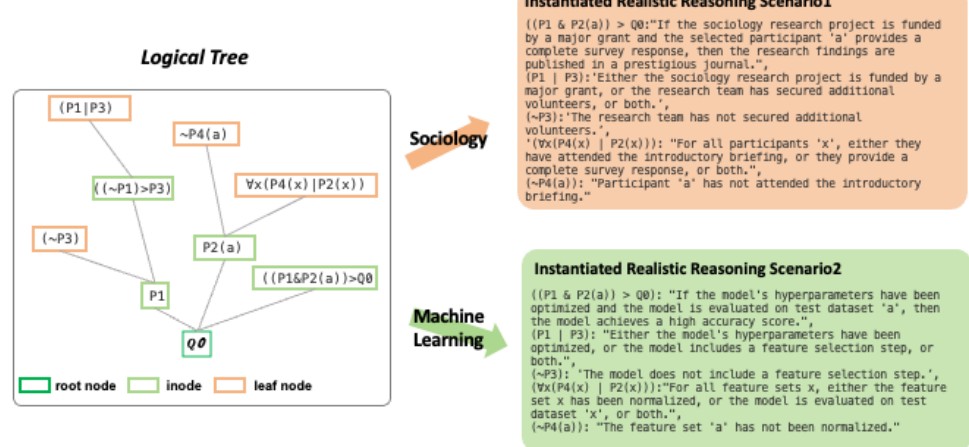

Figure 12: An example of instantiating a logic tree for diverse scenarios

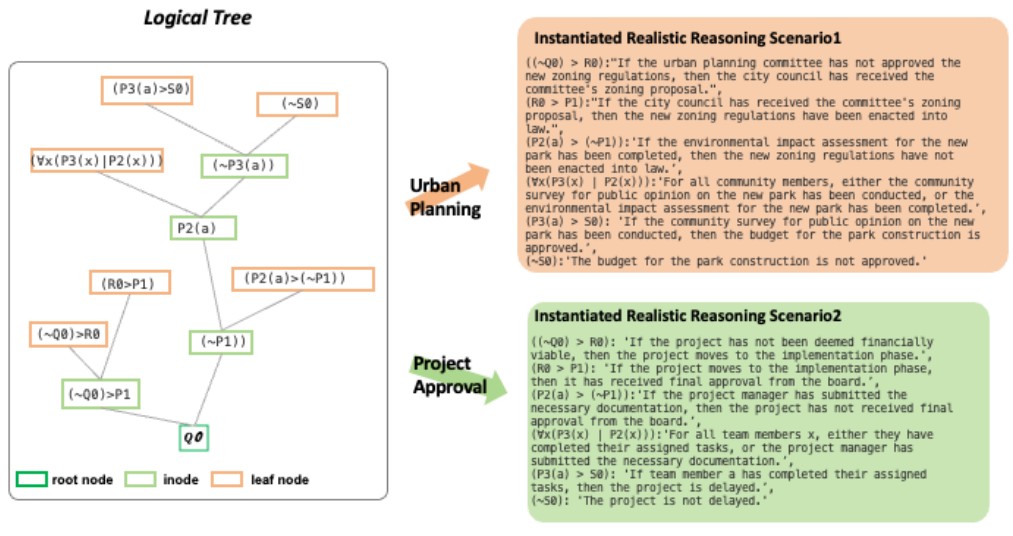

Figure 13: An example of instantiating a logic tree for diverse scenarios

