# OpenReview forum: "LogicTree: Improving Complex Reasoning of LLMs via Instantiated Multi-step Synthetic Logical Data"
_NeurIPS.cc/2025/Conference — NeurIPS 2025 spotlight_

### Official Review · Reviewer_G8Cd · 2025-06-21

**Clarity:** 2
**Significance:** 3
**Originality:** 3
**Rating:** 5
**Confidence:** 3

**Summary:**

This paper describes an approach to try and generate new synthetic data for logical inference (both propositional logic and first order logic are covered) with the goal of improving reasoning in LLMs.  The problem is an important one in the era of the so-called reasoning models.  The approach described here tries to define a 'LogicTree' which effectively describes a tree of rules applied using backward chaining, have an LLM assign realistic statements to each rule based on a specified domain, create a natural language reasoning process which presumably defines the steps in the reasoning process and then use that to fine tune a set of models.  Results are reported on a set of base models and it is shown that this benchmark training improves performance on other logic related datasets.

**Questions:**

Please see questions above.

**Ethical Concerns:**

["NO or VERY MINOR ethics concerns only"]

**Final Justification:**

I considered the authors' explanation of the core contribution and realized I had misunderstood it.

**Limitations:**

This section was outside of the 9 page limit, following references and had little substance.

**Paper Formatting Concerns:**

Limitations, implications are outside the main body of the paper.

**Quality:**

3

**Strengths And Weaknesses:**

Strengths:
1.  The approach is interesting, as is the creation of natural language versions of logical rules.
2.  The use of other logical benchmarks to see if the results generalize is also useful.

Weaknesses:
1.  I do not understand the attempt to define the LogicTree using some sort of 'pattern matching' structure matching approach - surely instrumenting any backward chaining symbolic system is likely to give you the same sort of tree?  Why re-invent what is well known and well implemented?
2.  Key details are lacking making the paper difficult to read, and hard to assess.
Examples:
a.  How were the initial problems specified?  Were they chosen from some existing logical benchmarks?
b.  How many such problems were there,
c.  What is the size/complexity of the problems - how many were propositional, and how many were first order logic.
d.  How did you assure that you had similar complexity problems to benchmarks from FOL theorem provers?
e.  How do you assure that when problems are indeed complex enough (such as those problems used by FOL theorem provers) the NL statements make sense at a semantic level?
3.  If the core contribution is the benchmark then the focus should be on benchmark characteristics and how they differ from other benchmarks.  If the core contribution is some sort of novel mechanism to teach LLMs reasoning the novelty was not clear.

Based on the author rebuttal I clearly misunderstood the main contribution here.  Thank you for the explanation - I will edit my scores appropriately.

---

> ### Author Rebuttal · Authors · 2025-07-31
>
> **Thank you for your valuable and constructive review. Below, we respond to each of your comments in detail and hope that our responses could properly address your concerns. If you find them satisfactory, we would be grateful if you could consider updating your score (Rating: 2: reject). Otherwise, please feel free to let us know any remaining questions or concerns, and we will be happy to continue the discussion.**
>
> # Motivation and Core Contributions of Our Work
> We sincerely apologize for any lack of clarity in our manuscript that may have caused confusion. To provide a clearer understanding of our work, we would like to begin by outlining the motivation and core contributions of this paper.
> ### 1. Motivation: The Scarcity of High-Quality, Multi-Step Logical Reasoning Data for LLM Training
>
> Our research is motivated by a critical challenge: **LLMs still exhibit significant weaknesses in complex, multi-step logical reasoning**. This limitaion underscore a significant scarcity of high-quality training data that embodies multi-step logical reasoning chains in natural language form.
>
> Prior work on synthesizing logical data suffers from key limitations: 1) **Limited Reasoning Complexity**, with limited combinations of logical rules.2) **Inadequate Real-World Instantiation**, lacking contextual coherence and fail to reflect realistic scenarios.
>
> ### 2. Our Main Contribution: The LogicTree Framework for Synthesizing Logical Reasoning Datasets for LLM Training
> We introduce LogicTree, **a novel framework for synthesizing multi-step logical reasoning datasets for LLM training**, offering advantages in terms of complexity and instantiation.  **The pipeline consists of two main stages**. First, we construct symbolic and diverse formal logic trees from scratch. Then, **we design a two-stage LLM-based method to instantiate these symbolic logic trees** into context-rich natural language narratives with diverse real-world scenarios.
>
> Our experiments show that **fine-tuning LLMs on the LogicTree-generated dataset** yields significant and consistent improvements across multiple challenging natural language logical reasoning benchmarks.
>
> # Weakness 1
> > ...any backward chaining symbolic system is likely to give you the same sort of tree? Why re-invent what is well known and well implemented?
>
> We appreciate the reviewer‘s observation regarding a crucial point that we wish to clarify:  the fundamental difference in purpose between LogicTree and backward-chaining symbolic systems. **Our work is not an attempt to reinvent backward chaining for proof verification, but rather to design a novel framework for data synthesis to train LLMs**.
> To address your question in detail, we organize our response along the following two aspects:
> ## 1. Key Difference in Core Objective: Synthesis, Not Verification
>
> ### 1.1 Backward chaining symbolic system for Verification
> Our understanding of these systems [1,2] are used for verification. Given a target conclusion or hypothesis, they **employ a backward-chaining mechanism to search a knowledge base for relevant facts and rules**. If a valid reasoning path is found, the system **outputs a grounded proof tree that supports the target conclusion**. For example:
>
> ```
>         Conclusion: is(charlie, cold)?
>                         |
> Rule 1. IF is(X, young) And is(X, round), THEN is(charlie, cold).
>                 /                 \
> Fact 1: is(Bob, young)         Fact 2: is(bob, round).
> ```
>
>
> **Backward-chaining symbolic systems are not well-suited for generating training data aimed at teaching LLMs logical reasoning**.
> 1. The process of searching for a reasoning path is deterministic and does not allow control over the number of reasoning steps, tree structure, or types of rules used.
> 2. The resulting proof tree is already instantiated — the validity of intermediate rules relies on specific factual knowledge. As a result, it lacks structural abstraction, making it unsuitable for synthesizing multiple diverse instantiations from the same underlying logic tree.
> ### 1.2 LogicTree for Data Synthesis
> In contrast, LogicTree is a synthesis framework that **starts from an arbitrary fully symbolic logical formula**. Guided by controllable specifications—such as the number of reasoning steps, rule types, and tree structure—it employs a backward-chaining strategy to **select applicable logical rules from a formal logic system**.
> ```
>         Logic formula: P -> Q
>                 | Rule: ((p → q)) ∧ (q → r)) ⊢ (p → r)
>              /      \
>  Premise1: P->T     Premise2: T->Q
> ```
> By instantiating abstract logic trees into diverse real-world scenarios, we aim to synthesize data that helps LLMs acquire fundamental logical reasoning skills.
> ### 1.3 Key Differences
> | |Backward chaining symbolic system |LogicTree|
> |-|-|-|
> |Inputs|A specific conclusion within a knowledge base|A fully symbolic logical expression|
> |Outputs|An instantiated proof path based on facts from the knowledge base|A structured, abstract logical reasoning tree|
> |Rules |A predefined set of rules in the knowledge base| General logical rules applied flexibly via pattern matching|
> |Objective|Verification: To find an existing proof that supports the conclusion|Synthesis: To construct new, structurally controllable logic from scratch|
>
> ## 2. Motivation for Structural Pattern Matching in Logical Formulas
> Under the synthesis objective, existing methods[3,4,5] for generating formal logic trees face a key limitation: **they require strict structural identity between formulas**, which severely limits the diversity of rule combinations.
>
> To overcome this, **we introduce Structural Pattern Matching—a more flexible backward deduction method that compares the structural skeletons of Abstract Syntax Trees** (ASTs, line 113). For example, consider the abstract rule of Hypothetical Syllogism, `HS: ((p→q)∧(q→r)) ⊢ (p→r)`. A traditional method requiring unification could only apply this to a simple `X→Y` conclusion and would fail with a more complex formula like `(A∨B)→(¬C)`. Our method enables us to apply the HS rule to decompose `(A∨B)→(¬C)` into two new sub-goals: `(A∨B)→G` and `G→(¬C)`.
>
> # Weakness 2
>
> Thank you for your valuable comments. We will respond to your questions point by point.
>
> ### Question 2.1
>
>  > a. How were the initial problems specified? Were they chosen from some existing logical benchmarks?
>
> We generate the entire logical reasoning tree from scratch based solely on fundamental logical rules, **without the need for any initial problems as input**. As discussed in more detail in Weakness 1.2 above, our reasoning tree generation starts from an arbitrary, fully symbolic logical expression — it is not derived from any real-world problem.
>
>
> ### Question 2.2
> >b. How many such problems were there,
> >c. What is the size/complexity of the problems.
>
> We generated 5k symbolic logic trees with **depths from 2 to 15**, and instantiated each into **3 semantically diverse scenarios**, yielding 1.5k reasoning problems. After applying an automatic verification process that discarded 8.73% of invalid samples, **the final dataset for LLM training contains **1.38k** high-quality, multi-step reasoning instances**.
>
>
> Due to space limitations, the table below presents statistical information for logic trees with 5 to 8 reasoning steps.
> | Step|Number of Nodes|Maximum Depth|Number of Distinct rules|Proportion of Fol|Proportion of Prop|
> |-|-|-|-|-|-|
> |5|$11.46_{\pm1.19}$|$3.68_{\pm1.42}$|$4.14_{\pm0.52}$|$31.42\%$|$68.58\%$|
> |6|$13.46_{\pm1.01}$|$5.04_{\pm0.32}$|$4.75_{\pm0.81}$|$30.45\%$|$69.55\%$|
> |7|$15.78_{\pm1.81}$|$5.61_{\pm0.56}$|$5.29_{\pm0.72}$|$29.94\%$|$70.06\%$|
> |8|$18.30_{\pm2.01}$|$6.04_{\pm0.76}$|$5.67_{\pm0.97}$|$28.82\%$|$71.18\%$|
>
>
> ### Questino 2.3
>
> > d. How did you assure that you had similar complexity problems to benchmarks from FOL theorem provers?
> > How do you ... the NL statements make sense at a semantic level?
>
> Since LLMs still face significant challenges in multi-step logical reasoning [6] — for example, their performance on 4-step and 5-step problems remains weak in Multi-LogiEval — **we focus on synthesizing problems that are currently difficult for LLMs to solve, in order to better support their learning**.
>
> We emphasize that our work is not intended as a benchmark for first-order logic provers, nor does it require alignment with their difficulty levels. Rather, **our objective is to generate logically valid, diverse, and controllable training samples that are suitable for improving LLM reasoning capabilities**.
>
> In our synthesis pipeline, we design a two-stage LLM-based approach to instantiate logic trees into realistic and coherent scenarios. The goal is not to have LLMs memorize fixed natural language statements, but rather to help them learn how fundamental logical reasoning rules are applied across diverse contexts. Extensive evaluations on multiple benchmarks demonstrate the effectiveness of our method.
>
>
> # Weakness 3
> > If the core contribution is ...
>
>
> As we stated at the beginning of our reply,  our work aims to address the scarcity of high-quality, multi-step logical reasoning data for LLM training. We propose LogicTree, a framework that synthesizes instantiated and structurally complex logical reasoning data. Such data is essential for teaching LLMs to develop robust, generalizable logical reasoning capabilities.
>
>
>
> ### Reference：
> [1] Lambada: Backward chaining for automated reasoning in natural language. ACL, 2023.
>
> [2] SymBa: Symbolic Backward Chaining for Structured Natural Language Reasoning. ACL, 2025.
>
> [3] Enhancing reasoning capabilities of llms via principled synthetic logic corpus. NIPS, 2024.
>
> [4] Learning deductive reasoning from synthetic corpus based on formal logic. ICML, 2023.
>
> [5] LogicAsker: Evaluating and improving the logical reasoning ability of large language models. EMNLP, 2024.
>
> [6] Multi-logieval: Towards evaluating multi-step logical reasoning ability of large language models. EMNLP, 2024.

---

> > ### Comment · Reviewer_G8Cd · 2025-08-03
> >
> > Thank you for the detailed explanation.  I clearly did misunderstand your key contributions - hence my earlier scores.  I am revising it now based on the contribution you described.

---

> > > ### Author Response · Authors · 2025-08-04
> > >
> > > Dear Reviewer G8Cd,
> > >
> > > Thank you very much for your encouraging and positive feedback on our paper. We are deeply grateful for your consideration that we have addressed your concerns. **We sincerely appreciate your careful re-evaluation of our work and your willingness to revise the score** after understanding our key contributions more clearly. Your comments have not only significantly improved the quality of our work but have also been a great source of encouragement for us.
> > >
> > > Thank you once again for your time and support.
> > >
> > > Best,
> > >
> > > Authors

---

### Official Review · Reviewer_xJSd · 2025-06-29

**Clarity:** 3
**Significance:** 2
**Originality:** 3
**Rating:** 5
**Confidence:** 3

**Summary:**

This paper introduces LogicTree, a framework for synthesizing multi-step logical reasoning datasets designed to enhance the complex reasoning capabilities of large language models (LLMs). The core idea involves generating symbolic logical reasoning trees via backward deduction using structural pattern matching, and then instantiating these trees into natural language using a two-stage LLM-based prompting strategy. This approach enables the construction of realistic, context-rich scenarios that support generalizable reasoning. The authors demonstrate that LogicTree outperforms several baselines across multiple logical reasoning benchmarks, particularly in multi-step reasoning tasks, achieving an average improvement of 10.5% in accuracy.

**Questions:**

Q1. The paper emphasizes the effectiveness of the method on smaller models, but then why are 1B or 3B-scale models not evaluated?

Q2. Given that recent advances in long-chain Chain-of-Thought (CoT) prompting have significantly improved reasoning capabilities in language models, have you conducted any experiments or comparisons related to this line of work?

**Ethical Concerns:**

["NO or VERY MINOR ethics concerns only"]

**Final Justification:**

The rebuttal addressed my concerns. No issues remain unresolved.
However, the authors supplemented a significant amount of experiments during the rebuttal phase—experiments that should have been part of the original submission. I am uncertain whether this is permissible.

**Limitations:**

yes

**Paper Formatting Concerns:**

No.

**Quality:**

3

**Strengths And Weaknesses:**

Strengths:

Quality: The paper presents a well-designed methodology that combines symbolic logic with modern LLMs for dataset synthesis. The experimental results are comprehensive and show clear improvements over strong baselines.

Clarity: The writing is clear, with helpful diagrams and algorithm pseudocode.

Significance: LogicTree contributes meaningfully to enhancing logical reasoning abilities in LLMs, particularly in complex, multi-step settings.

Originality: While the use of tree structures for logical reasoning is not new and has been employed in prior works such as proof trees, symbolic reasoning chains, and MCTS-based methods, this paper distinguishes itself through its combination of structural pattern matching-based logic tree generation and a two-stage LLM-driven instantiation process.

Weaknesses:

Quality: While the paper provides strong empirical evidence of LogicTree’s effectiveness on multi-step reasoning, it lacks a formal analysis of algorithmic complexity (e.g., runtime or space complexity) for logic tree generation or instantiation. There is no statistical breakdown of the structural complexity of the generated logic trees (e.g., average depth, node count, rule variety), which would help quantify dataset diversity and difficulty. The paper includes Qwen2.5-7B results only in Multi-LogiEval, and omits it from the main result table (Table 1) and other benchmark evaluations, limiting the completeness of cross-model comparison.

Clarity: Some table captions do not totally match with the content or the context around.

Significance: While LogicTree achieves significant performance gains on smaller models (e.g., +10.5% for LLaMA3-8B), the improvement becomes marginal on stronger models like LLaMA3-70B (+2.3%). The paper devotes substantial space to analyzing the weakest model, but lacks a discussion on why stronger models benefit less. This raises concerns about the long-term utility and scalability of the method, especially as modern 8B-class models (e.g., Qwen3-8B, DeepSeek-VL) continue to improve.

---

> ### Author Rebuttal · Authors · 2025-07-31
>
> **Thank you for your valuable and constructive review. We sincerely appreciate your feedback. Below, we respond to each of your comments in detail and hope that our responses could properly address your concerns. If you find them satisfactory, we would be grateful if you could consider updating your score. Otherwise, please feel free to let us know any remaining questions or concerns, and we will be happy to continue the discussion.**
>
> # Weakness 1
> ### Weakness 1.1
> > it lacks a formal analysis of algorithmic complexity (e.g., runtime or space complexity) for logic tree generation or instantiation.
>
> We appreciate your attention to the computational complexity of our data synthesis process, which is indeed crucial for the practical applicability of our method. Below, we provide a detailed analysis of the algorithmic complexity for both logic tree generation and LLM-based instantiation:
>
> * **Logic tree generation**: As illustrated in Algorithm 1, the process of generating a symbolic reasoning tree has a **time complexity of $O(NM)$**, where N is the number of reasoning steps and M is the number of available logical rules. **Since both N(<20) and M(<200) are relatively small in practice, the computational cost of this step is negligible**.
> * LLM-based instantiation: The computational cost of calling LLMs is typically measured by **the number of output tokens they generate**, which serves as a practical indicator of runtime and resource usage. Specifically: 1) The first call to LLM populates the abstract logical symbols with concrete natural language entities. We denote the number of output tokens consumed in this step as $c$. 2) The second call translates the sequence of reasoning steps into natural language. We assume that translating each individual reasoning step requires an average of $T$ output tokens. The computational cost of generating a data instance with reasoning steps $m$ is:
>   $$Cost_{inst}=m⋅T+c$$
> For a more detailed analysis of the complexity of invoking LLMs, please refer to Appendix B.5.
>
> Our framework guarantees correctness by design, thereby eliminating the need for multiple LLM calls. As a result, the generation cost remains acceptable compared to other approaches.
>
> ### Weakness 1.2
> >  There is no statistical breakdown of the structural complexity of the generated logic trees (e.g., average depth, node count, rule variety),
>
> We sincerely thank you for your insightful comment and apologize for the lack of this information in the previous version. We provide the relevant details below.
> The table below presents statistical information for **logic trees with 5 to 8 reasoning steps**, including (1) the average maximum depth of the trees, (2) the average number of nodes per tree, (3) the average number of distinct rules applied, as well as (4) the proportion of first-order logic rules and propositional logic rules used in each tree.
>
>
>
> |Step|Number of Nodes|Maximum Depth|Number of Distinct rules|Proportion of Fol|Proportion of Prop|
> |-|-|-|-|-|-|
> |5|$11.46_{\pm1.19}$|$3.68_{\pm1.42}$|$4.14_{\pm0.52}$|$31.42\%$|$68.58\%$|
> |6|$13.46_{\pm1.01}$|$5.04_{\pm0.32}$|$4.75_{\pm0.81}$|$30.45\%$|$69.55\%$|
> |7|$15.78_{\pm1.81}$|$5.61_{\pm0.56}$|$5.29_{\pm0.72}$|$29.94\%$|$70.06\%$|
> |8|$18.30_{\pm2.01}$|$6.04_{\pm0.76}$|$5.67_{\pm0.97}$| $28.82\%$|$71.18\%$|
>
> ### Weakness 1.3
> > The paper includes Qwen2.5-7B results only in Multi-LogiEval, and omits it from the main result table (Table 1) and other benchmark evaluations, limiting the completeness of cross-model comparison.
>
> Thank you for your valuable comments. We apologize for omitting this part of the evaluation due to space limitations in the main text. We have now added the experimental results of the Qwen2.5-7B model as follows.
>
> |Qwen2.5-7B|LogicBench|LogiQA2.0|FOLIO|BBH-Logical|AGIEval-LR|AGIEval-AR|Avg|
> |-|-|-|-|-|-|-|-|
> |Vanilla|83.1|63.8|54.3|72.8|65.3|23.9|60.4|
> |+PAPARULE|82.9|**65.1**|55.3|72.8|68.3|24.5|61.5|
> |+LogicAsker|84.4|64.9|58.7|72.8|66.7|24.5|62.1|
> |+FLD$_{\times 2}$|83.4|**65.1**|59.6|73.6|67.3|26.1|62.5|
> |LogicTree|**89.2(+6.1)**|$\underline{64.9}$(+1.1)|**63.8(+9.5)**|**74.3(+1.5)**|**69.3(+4.0)**|**28.7(+4.8)**|65.0|
>
> As shown in the table, LogicTree demonstrates consistent and comprehensive performance improvements on Qwen2.5-7B, in line with our main experimental results. On average, it achieves a 4.6% increase in accuracy across multiple benchmarks. Notably, it achieves gains of 6.1% on LogicBench and 9.5% on FOLIO, both of which require more advanced logical reasoning capabilities.
> # Weakness 2
> > Some table captions do not totally match with the content or the context around.
>
> Thank you for your careful review and valuable comments. We will thoroughly check the formatting and content throughout the paper and make the necessary corrections in the final version.
> # Weakness 3
> > The paper devotes substantial space to analyzing the weakest model, but lacks a discussion on why stronger models benefit less...
>
> Thank you for your valuable comments. First, we would like to clarify that the average improvement for LLaMA3.1-70B is relatively modest (2.3%). This is primarily because LLaMA3.1-70B already achieves high scores on several relatively simple benchmarks, leaving limited room for further improvement. It is worth noting that the LLaMA3.1-70B model still achieves substantial improvements on several challenging benchmarks, such as a **12.1%** gain on MultiLogic-Eval(Appendix Table 5) and a **5.6%** gain on FOLIO.
>
> To further address your concerns, we conducted additional experiments using the well-known **DeepSeek series of distilled models**.
>
> |DeepSeek-R1-Distill-Qwen-7B|LogicBench|LogiQA2.0|FOLIO|AGIEval-LR|AGIEval-AR|ML-D1|ML-D2|ML-D3|ML-D4|ML-D5|Avg|
> |-|-|-|-|-|-|-|-|-|-|-|-|
> |Vanilla|86.8|52.4|61.2|56.8|32.6|70.0|57.0|60.0|55.0|44.4|57.6|
> |+LogicTree|90.6(+3.8)|53.4(+1.0)|63.4(2.2)|62.4(5.6)|34.4(+1.8)|80.9(+10.9)|67.6(+10.6)|62.5(+2.5)|56.6(+1.6)|53.3(+9.9)|**62.5(+4.9)**|
>
> |DeepSeek-R1-Distill-Llama-8B|LogicBench|LogiQA2.0|FOLIO|AGIEval-LR|AGIEval-AR|ML-D1|ML-D2|ML-D3|ML-D4|ML-D5|Avg|
> |-|-|-|-|-|-|-|-|-|-|-|-|
> |Vanilla|83.1|53.4|53.1|53.3|30.4|78.0|67.1|60.0|49.2|46.6|57.4|
> |+LogicTree|91.2(+8.1)|54.9(+1.5)|57.8(4.7)|57.3(+4.0)|33.9(+3.5)|86.3(+7.3)|73.3(+6.2)|66.0(+6.0)|62.5(+13.3)|56.6(+10.0)|**63.9(+6.5)**|
>
> As shown in the table, LogicTree consistently improves the performance of DeepSeek-R1-Distill-Llama-8B and DeepSeek-R1-Distill-Qwen-7B across multiple benchmarks, with average gains of **6.5%** and **4.9%**, respectively.
> # Question 1
> > The paper emphasizes the effectiveness of the method on smaller models, but then why are 1B or 3B-scale models not evaluated?
>
> Thank you for your valuable comments. We have conducted additional experiments using Qwen2.5-1.5B and Qwen2.5-3B to further evaluate the effectiveness of LogicTree on smaller language models.
>
>
> |Qwen2.5-1.5B|LogicBench|LogiQA2.0|FOLIO|AGIEval-LR|AGIEval-AR|ML-D1|ML-D2|ML-D3|ML-D4|ML-D5|Avg|
> |-|-|-|-|-|-|-|-|-|-|-|-|
> |Vanilla|65.0|43.9|48.5|39.5|17.4|65.6|53.3|48.8|43.3|31.1|45.8|
> |+LogicTree|74.4|48.4|53.4|42.4|21.3|80.9|56.2|54.8|54.2|44.4|**53.5(+7.7)**|
>
> |Qwen2.5-3B|LogicBench|LogiQA2.0|FOLIO|AGIEval-LR|AGIEval-AR|ML-D1|ML-D2|ML-D3|ML-D4|ML-D5|Avg|
> |-|-|-|-|-|-|-|-|-|-|-|-|
> |Vanilla|74.4|53.9|51.0|53.5|20.9|74.5|55.2|46.5|45.0|36.6|51.2|
> |+LogicTree|83.9|56.2|61.2|55.7|24.4|86.3|67.6|59.3|58.3|50.0|**60.5(+9.3)**|
>
> As shown in the table, LogicTree effectively enhances the logical reasoning capabilities of smaller models such as Qwen2.5-1.5B and Qwen2.5-3B, achieving consistent and substantial improvements across multiple benchmarks. Specifically, Qwen2.5-1.5B achieves an average accuracy gain of 7.7%, and Qwen2.5-3B improves by 9.3%, demonstrating that our synthesized data remains effective even for smaller-scale models.
> # Question 2
> > Given that recent advances in long-chain Chain-of-Thought (CoT) prompting ..., have you conducted any experiments or comparisons related to this line of work?
>
> We appreciate you pointing out this critical issue—it indeed relates to one of our core contributions. In fact, we provide **a detailed discussion of this aspect in Appendix A.1**. We apologize for not highlighting it more clearly in the main text, which may have caused confusion. To address this, we will add an appropriate reference in Section 5.
>
> **For your convenience, we summarize the key differences below**:
>
> Many efforts focus on designing more effective prompting strategies to help LLMs complete complex logical reasoning tasks, such as Symbolic Chain-of-Thought (SymbCoT)[1]  and Logic-of-Thought[2]. Although prompt-based methods can effectively leverage the potential of large language models(LLMs), their performance remains fundamentally constrained by the models’ inherent reasoning capabilities.
>
> **These works aim to elicit the existing reasoning capabilities of LLMs through carefully designed prompts, rather than enhancing their intrinsic reasoning abilities**. In contrast, our approach focuses on synthesizing a large volume of high-quality logical reasoning data from a training perspective, which helps to strengthen the inherent reasoning skills of LLMs. **Moreover, our method is not in conflict with these prompt-based approaches — rather, they are complementary. We believe that developing a unified framework that bridges training-based and prompt-based methods is a promising direction for future research**.
>
> We humbly hope our response has addressed your concerns. If you have any additional concerns or comments that we may have missed in our responses, we would be most grateful for any further feedback from you to help us further enhance our work.
>
> ## Reference:
> [1] Faithful logical reasoning via symbolic chain-of-thought. ACL, 2024.
>
> [2] Logic-of-thought: Injecting logic into contexts for full reasoning in large language models. NAACL,2024

---

> > ### Comment · Reviewer_xJSd · 2025-08-02
> >
> > Thank you for your reply. These experiments should have been included when you first submitted the initial version, not added in rebuttal. However, the results seem acceptable, so I'll raise your score by one point.

---

> > > ### Author Response · Authors · 2025-08-02
> > >
> > > Dear Reviewer xJSd,
> > >
> > > Thank you very much for your encouraging and positive feedback on our paper. We are deeply grateful for your consideration that we have addressed your concerns. **We understand your point that these experiments should ideally have been included in the initial submission, and we truly appreciate your willingness to still re-evaluate our work based on the updated results**. Your thoughtful evaluation and constructive feedback are invaluable to us, and **we sincerely appreciate your decision to raise the overall score based on our response**.
> > >
> > > Thank you once again for your time and support.
> > >
> > > Best,
> > >
> > > Authors

---

### Official Review · Reviewer_sD1m · 2025-06-30

**Clarity:** 2
**Significance:** 3
**Originality:** 3
**Rating:** 5
**Confidence:** 4

**Summary:**

The paper proposes a novel framework called LogicTree for synthesizing multi-step logical reasoning datasets. The method leverages first-order logic rules to generate complex reasoning trees and instantiates them using Large Language Models to produce natural language reasoning data with realistic scenarios. The paper conducts experiments across multiple benchmarks and demonstrates a good improvement of accuracy on complex logical reasoning tasks.

**Questions:**

1. Can you provide more detailed statistics on the generated datasets?
2. How does the LLM-based instantiation process handle the hallucination issues of LLMs to ensure the correctness of the generated statements?
3. Can you consider using more advanced large reasoning models as DeepSeek-R1-distill-8B, to further validate the effectiveness of the LogicTree framework in improving the reasoning abilities of LLMs?

**Ethical Concerns:**

["NO or VERY MINOR ethics concerns only"]

**Final Justification:**

The responses of the author address my concerns about the data statistics, quality control, and performance improvement for advanced LRM. Thus, I update my score to accept.

**Quality:**

3

**Strengths And Weaknesses:**

## Paper Strengths:
1. The proposed LogicTree framework is innovative in efficiently synthesizing multi-step logical reasoning datasets with complex reasoning patterns.
2. The two-stage LLM-based instantiation technique for translating symbolic logical expressions into concrete statements with contextual significance is insightful. This approach enhances the realism and applicability of the generated datasets.
3. The paper provides in-depth experimental results demonstrating the effectiveness of LogicTree in enhancing LLMs' reasoning abilities across various benchmarks and questions of different complexity levels.

## Paper Weaknesses:
1. Lack of details statistics on the generated datasets, such as the number of questions, unique reasoning patterns, complexity levels, and distribution of question types. This information is crucial for understanding the diversity and coverage of the synthesized datasets.
2. The LLM-based instantiation purely relies on the LLMs' capabilities to generate natural language statements and inject factual knowledge. However, the LLMs suffer from hallucination issues, which may lead to the generation of incorrect or misleading statements with unreliable factual knowledge. The paper should provide more analysis on how the instantiation process handles such issues and ensures the correctness of the generated statements.
3. The used baseline LLMs are just the vanilla LLMs without being specifically designed for intricate reasoning abilities. The paper should consider using more advanced large reasoning models, such as DeepSeek-R1-distill-8B, to further validate the effectiveness of the LogicTree framework in improving the reasoning abilities of LLMs.

---

> ### Author Rebuttal · Authors · 2025-07-31
>
> Thank you for your valuable and constructive review. We sincerely appreciate your feedback. Below, we respond to each of your comments in detail and hope that our responses could properly address your concerns. If you find them satisfactory, we would be grateful if you could consider updating your score. Otherwise, please feel free to let us know any remaining questions or concerns, and we will be happy to continue the discussion.
>
> # Weakness 1 & Question 1
> > Lack of details statistics on the generated datasets, such as the number of questions, unique reasoning patterns, complexity levels, and distribution of question types.
>
> Thank you for your valuable comments. We generated 5,000 symbolic logic trees with **depths from 2 to 15**, and instantiated each into **3 semantically diverse scenarios**, yielding 15,000 reasoning problems. After applying an automatic filtering process that discarded 8.73% of noisy or invalid samples, **the final dataset for LLM training contains **1.38k** high-quality, multi-step reasoning instances**.
>
> An analysis of our synthetic data's properties and a comparison with other methods are presented in **Appendix C.3**. For the reviewer's convenience, we reproduce the key information below.
> ||Logic Rules|Reasoning Steps|Translation|Instantiation|
> |-|-|-|-|-|
> |RuleTaker|2|1-5|Template|Random Entities|
> |PARARULE|2|1-5 |Template|Random Entities|
> |FLD|13|1-8 |Template|WorldNet |
> |FLD$_{\times 2}$|$\approx$ 50|1-8|Template|WorldNet|
> |**LogicTree**|**190**|**1-15**|**LLM-based**|**Realistic Scenario**|
>
> In addition, the following table presents supplementary statistical information to further characterize our dataset. The table below presents statistical information for **logic trees with 5 to 8 reasoning steps**, including (1) the average maximum depth of the trees, (2) the average number of nodes per tree, (3) the average number of distinct rules applied, as well as (4) the proportion of first-order logic rules and propositional logic rules used in each tree.
>
>
>
> |Step|Number of Nodes|Maximum Depth|Number of Distinct rules|Proportion of Fol|Proportion of Prop|
> |-|-|-|-|-|-|
> |5|$11.46_{\pm1.19}$|$3.68_{\pm1.42}$|$4.14_{\pm0.52}$|$31.42\%$|$68.58\%$|
> |6|$13.46_{\pm1.01}$|$5.04_{\pm0.32}$|$4.75_{\pm0.81}$|$30.45\%$|$69.55\%$|
> |7|$15.78_{\pm1.81}$|$5.61_{\pm0.56}$|$5.29_{\pm0.72}$|$29.94\%$|$70.06\%$|
> |8|$18.30_{\pm2.01}$|$6.04_{\pm0.76}$|$5.67_{\pm0.97}$| $28.82\%$|$71.18\%$|
>
>
> # Weakness 2 & Question 2
> > How does the LLM-based instantiation process handle the hallucination issues of LLMs to ensure the correctness of the generated statements?
>
> Thank you for this insightful question. Ensuring the correctness of the generated statements is a critical component of our methodology. We address this concern from two key perspectives: (1) our proactive methodological design and (2) a systematic error verification process.
> * **Two-stage LLM-based instantiation**: Due to the inherent limitations of LLMs in handling complex logical reasoning, directly prompting them to generate realistic logical reasoning problems from symbolic logic trees may introduce errors and hallucinations. Therefore, to address this challenge, **we designed a two-stage prompting strategy to reduce the task's difficulty for the LLM**. The first stage prompts the model to assign factually and logically coherent entities to the symbolic variables in the leaf nodes. Subsequently, the second stage instructs the LLM to translate the symbolic reasoning process into natural language on a step-by-step basis. This decomposition ensures that when generating the natural language statements, **the LLM only needs to focus on the translation of individual logical expressions**, rather than grappling with the complex dependencies of the entire logic tree. This targeted approach effectively minimizes hallucinations and improves the fidelity of the generated natural language statements.
> * **Error verification process**：Furthermore, to minimize errors in the instantiated data, we implemented a systematic verification process. For each generated instance, **we employed an LLM-based verifier to check for logical fallacies and implausible natural language statements** by comparing them against their corresponding logical expressions. This process successfully **filtered out 8.73% of erroneous data**, thereby significantly enhancing the overall quality of the dataset.
>
> To further address the reviewer's concern, we manually evaluated 500 randomly selected samples and found an error rate of only 4.6% (23/500). These minor errors were primarily caused by the model using incorrect entities or misunderstanding certain logical relations within a single reasoning step.
>
> Moreover, we would like to emphasize that the purpose of our synthetic data is **not to** teach the LLM to memorize specific instantiations of natural language statements, but rather to **help it learn to apply underlying logical rules across diverse instantiation scenarios**. Given this objective, we contend that this small proportion of noise is acceptable.
>
>
> # Weakness 3 & Question 3
>
> > Can you consider using more advanced large reasoning models as DeepSeek-R1-distill-8B, to further validate the effectiveness of the LogicTree framework in improving the reasoning abilities of LLMs?
>
> We appreciate your suggestion to supplement our work with experiments involving more advanced reasoning models, as this would provide important evidence to further validate the effectiveness of our dataset. We further include experiments using advanced reasoning models such as DeepSeek-R1-Distill-Llama-8B and DeepSeek-R1-Distill-Qwen-7B to strengthen our evaluation.
>
> |DeepSeek-R1-Distill-Qwen-7B|LogicBench|LogiQA2.0|FOLIO|AGIEval-LR|AGIEval-AR|ML-D1|ML-D2|ML-D3|ML-D4|ML-D5|Avg|
> |-|-|-|-|-|-|-|-|-|-|-|-|
> |Vanilla|86.8|52.4|61.2|56.8|32.6|70.0|57.0|60.0|55.0|44.4|57.6|
> |+LogicTree|90.6(+3.8)|53.4(+1.0)|63.4(2.2)|62.4(5.6)|34.4(+1.8)|80.9(+10.9)|67.6(+10.6)|62.5(+2.5)|56.6(+1.6)|53.3(+9.9)|**62.5(+4.9)**|
>
> |DeepSeek-R1-Distill-Llama-8B|LogicBench|LogiQA2.0|FOLIO|AGIEval-LR|AGIEval-AR|ML-D1|ML-D2|ML-D3|ML-D4|ML-D5|Avg|
> |-|-|-|-|-|-|-|-|-|-|-|-|
> |Vanilla|83.1|53.4|53.1|53.3|30.4|78.0|67.1|60.0|49.2|46.6|57.4|
> |+LogicTree|91.2(+8.1)|54.9(+1.5)|57.8(4.7)|57.3(+4.0)|33.9(+3.5)|86.3(+7.3)|73.3(+6.2)|66.0(+6.0)|62.5(+13.3)|56.6(+10.0)|**63.9(+6.5)**|
>
> As shown in the table, LogicTree consistently improves the performance of DeepSeek-R1-Distill-Llama-8B and DeepSeek-R1-Distill-Qwen-7B across multiple benchmarks, with average gains of 6.5% and 4.9%, respectively. Notably, on logic-intensive benchmarks such as the ML series, the improvements reach up to 13.3% and 10.9%. These results further confirm the effectiveness of LogicTree in enhancing the logical reasoning capabilities of LLMs.
>
> We humbly hope our response has addressed your concerns. If you have any additional concerns or comments that we may have missed in our responses, we would be most grateful for any further feedback from you to help us further enhance our work.

---

> > ### Comment · Reviewer_sD1m · 2025-08-01
> >
> > Thanks for the response, which addressed my concerns. I'll update my scores accordingly.

---

> > > ### Author Response · Authors · 2025-08-01
> > >
> > > Dear Reviewer sD1m,
> > >
> > >
> > > Thank you very much for your encouraging and positive feedback on our paper. We are deeply grateful for your consideration that we have **addressed your concerns**. Your thoughtful evaluation and constructive feedback are invaluable to us, and **we sincerely appreciate your willingness to consider raising your overall score based on our response**.
> > >
> > > Thank you once again for your time and support.
> > >
> > > Best,
> > >
> > > Authors

---

### Official Review · Reviewer_rqDh · 2025-07-03

**Clarity:** 3
**Significance:** 3
**Originality:** 2
**Rating:** 5
**Confidence:** 3

**Summary:**

The authors introduce “LogicTree,” a framework for synthesizing multi-step logical reasoning datasets with a focus on complexity and real-world contextual instantiation. The approach uses structural pattern matching for backward deduction to generate logic trees governed by first-order logic rules, and then employs a two-stage LLM-based method to instantiate these symbolic trees into diverse, context-rich, natural language scenarios. Logical consistency is checked post-hoc to filter poor samples. Experiments across several logical reasoning benchmarks demonstrate that training LLMs with LogicTree data improves accuracy over baselines.

**Questions:**

* What are the practical, computational costs of large-scale synthetic data generation and verification using LLMs, and do these limit applicability for broader academic use?
* How sensitive are the results to the number of scenario instantiations per logic tree? Is there an upper bound after which gains saturate or reverse?

**Ethical Concerns:**

["NO or VERY MINOR ethics concerns only"]

**Final Justification:**

The responses address most of my concerns about the LLM verification and more detailed analysis of the dataset. Detailed statistics here should be added to the paper to make it clearer for readers.

**Limitations:**

Yes

**Quality:**

3

**Strengths And Weaknesses:**

**Strengths**
* Methodological soundness: The paper clearly articulates the LogicTree pipeline, which systematically combines symbolic backward deduction with LLM-based scenario instantiation. The logic tree generation based on structural pattern matching is well-grounded in formal logic and illustrated stepwise.
* Comprehensive evaluation: The experiments benchmark the method across diverse challenging datasets and tasks, including ablation studies for scenario instantiation and process diversity. Results support the main claims with quantitative improvements.

**Weaknesses**
* LLM-based verification: The logic consistency verification relies on LLMs themselves to make sure consistency between instantiated text and logic formulas (Section 3.3). This may introduces a potential failure mode where systematic instantiation errors might go undetected.
* Depth in real-world evaluation: While the benchmarks are diverse and the accuracy improvements are robust, there is insufficient qualitative analysis to illustrate how the reasoning behavior of LLMs undergoes qualitative changes after training with LogicTree. For instance, do the models reason more explicitly, avoid hallucinations, or perform better in out-of-domain real-world reasoning? Specifically, on which types of questions are the improvements most significant, and on which do they even decline? More interpretive or failure-case analyses would strengthen the argument that such improvements stem from generalization rather than mere memorization.

---

> ### Author Rebuttal · Authors · 2025-07-31
>
> **We thank the reviewer for the insightful and valuable comments. We respond to each comment as follows and sincerely hope that our rebuttal could properly address your concerns. If so, we would deeply appreciate it if you could raise your score (Rating: 4: Borderline accept). If not, please let us know your further concerns, and we will continue actively responding to your comments and improving our submission**.
> ## Weakness 1
> > LLM-based verification: ...This may introduces a potential failure mode where systematic instantiation errors might go undetected.
>
> Thank you for your valuable comment. We acknowledge that using LLMs for verification inevitably introduces some detection errors. However, **our framework mitigates this by simplifying the verification task**. Instead of tasking the LLM with the complex evaluation of global logical consistency, **we decompose the problem into a series of localized verification steps**. In each step, the LLM simply verifies the translation between a logical expression and its natural language counterpart, a task at which state-of-the-art models excel [1,2]. **Once each step is correctly translated, the symbolic nature of the LogicTree inherently ensures the overall logical consistency**.
>
> **To further address the reviewer's concern, we manually evaluated 500 randomly selected samples and found an error rate of only 4.6% (23/500)**. These minor errors were primarily caused by the model using incorrect entities or misunderstanding certain logical relations within a single reasoning step.
>
> Moreover, we would like to emphasize that the purpose of our synthetic data is **not to teach the LLM to memorize specific instantiations of natural language statements, but rather to help it learn to apply underlying logical rules across diverse instantiation scenarios**. Given this objective, we contend that this small proportion of noise is acceptable.
>
> ## Weakness 2
> > Depth in real-world evaluation: ...More interpretive or failure-case analyses would strengthen the argument that such improvements stem from generalization rather than mere memorization.
>
> We sincerely appreciate your insightful observation that more in-depth qualitative analysis is necessary to substantiate our quantitative findings. We will include more examples of real test cases in the appendix of the final version. Here, we present a representative examples, obtained by **comparing the performance of Llama-3.1-8B-Vanilla and our LogicTree-trained model on the MultiLogic-Eval benchmark**.
>
>
> ### Case 1
> ```
> Premise 1 : If a biological sample is labeled "High-Risk" (P), then the handler must wear a Level-A protective suit (Q). (P → Q )
>
> Premise 2 : A review of today's lab footage shows that a researcher, Dr. Li, was not wearing a Level-A protective suit while handling sample S-007.(¬Q)
>
> Conclusion to Evaluate: an it be determined that sample S-007 was labeled "High-Risk"? (¬P)
> ```
>
> Vanilla Model's Response: No
>
> LogicTree-trained Model's Response: Yes
>
> **The LogicTree-trained model correctly identifies and applies Modus Tollens (a core idea underlying the well-known method of proof by contradiction)**. It adheres closely to the given logical structure, avoiding distractions from irrelevant speculations. This indicates that the training process has effectively internalized fundamental deductive principles, allowing the model to reason reliably from premises rather than relying on superficial pattern matching.**earning this rule is instrumental in enabling the model to apply proof by contradiction—an essential technique in mathematical reasoning.**
>
> ## Question 1
> > What are the practical, computational costs of large-scale synthetic data generation and verification using LLMs
>
> We appreciate the reviewer's insightful comment regarding the importance of computational cost associated with synthetic data generation. Indeed, a detailed analysis of the generation cost is provided in Appendix.B.5. For the reviewer's convenience, we summarize the key findings below. To clarify our computational cost, we analyzed *the number of LLM output tokens* consumed when generating each data instance.
>
> * **Instantiation Cost**:  1) The first call to LLM populates the abstract logical symbols with concrete natural language entities. We denote the number of output tokens consumed in this step as $c$. 2) The second call translates the sequence of reasoning steps into natural language. We assume that translating each individual reasoning step requires an average of $T$ output tokens. The computational cost of generating a data instance with reasoning steps $m$ is:
> $$Cost_{inst}=m⋅T+c$$
> * **Verification Cost**: We assume that the complexity of verifying a single step is comparable to that of translating it during instantiation. Consequently, the computational cost for verifying a data instance with m reasoning steps is:
> $$Cost_{ver}=m⋅T$$
>
> ||number of LLM invocations|Number of consumed LLM output tokens|Number of synthesized tokens|explanation
> |-|-|-|-|-|
> |V-STaR[1]|k|mkT|mT|k is the reasoning times.|
> |ALPHALLM[2]|mbk|$mbk(T+T_{rollout})$|mT|b is the reasoning times per step, k is the number of rollout simulations for each node, and $T_{rollout}$ is the additional tokens consumed during the rollout process.|
> |**LogicTree**|3|2mT+c|mT| |
>
> As shown in the table, the ratio of **synthesized tokens to consumed tokens** for our method is $\frac{mT}{2mT+c}$ (close to 1/2,in our data $T = 79_{\pm14}$, $c = 114_{\pm44}$),which is significantly lower than the cost of other methods.   Our framework's guarantee of correctness eliminates the need for multiple LLM calls. Consequently, its generation cost is entirely acceptable in comparison with other approaches.
>
>
> ## Question 2
> > How sensitive are the results to the number of scenario instantiations per logic tree? Is there an upper bound after which gains saturate or reverse?
>
> We sincerely thank the reviewer for the insightful comment. We would like to clarify that an initial analysis on this topic was provided in our ablation study in **Section 4.4** of the manuscript.
>
> In that section, we instantiated each logical tree into a varying number of reasoning scenarios (from 1 to 3), with our main results using three scenarios. As presented in Table 2, we observed that increasing the number of scenarios led to a consistent performance improvement across most benchmarks (e.g., from 51.6% to 55.3%). This finding supports our conclusion that greater diversity in reasoning scenarios is beneficial for the model to learn genuinely generalizable logical reasoning skills.
>
> We apologize for the omission and agree that exploring the upper bounds of this effect is valuable. To address this, we have conducted additional experiments for 4 and 5 instantiated scenarios. The complete set of results, including these new data points, is summarized below:
>
>
> |Model|LogicBench|LogiQA2.0|BBH|AGIEval-LR|AGIEval-AR|Avg|
> |-|-|-|-|-|-|-|
> |Llama3.1-8B|80.0|42.4|39.3|48.4|20.7|45.0|
> |num=1|83.1|51.2|49.6|54.1|23.9|51.6|
> |num=2|88.1|52.9|50.8|52.9|26.1|53.2|
> |num=3(used in LogicTree)|90.6|**53.5**|**53.2**|**56.1**|**26.5**|**55.3**|
> |num=4|92.1|52.7|**53.2**|**56.1**|25.7|55.2|
> |num=5|**94.5**|50.3|50.8|51.2|23.7|54.1|
>
> **As the results in the table indicate, the performance of Llama-3.1-8B peaks when using 3 or 4 instantiations**. However, upon further increasing the number to 5, we observe a slight decline in performance. Therefore, **taking into account both the computational overhead and the performance curve, using 3 instantiations offers the best cost-benefit trade-off and was chosen for our main experiments**.
>
> We humbly hope our response has addressed your concerns. If you have any additional concerns or comments that we may have missed in our responses, we would be most grateful for any further feedback from you to help us further enhance our work.
>
> ### Reference:
> [1] Faithful Logical Reasoning via Symbolic Chain-of-Thought. ACL, 2024
>
> [2] Divide and Translate: Compositional First-Order Logic Translation and Verification for Complex Logical Reasoning. ICLR, 2025

---

> > ### Comment · Reviewer_rqDh · 2025-08-05
> >
> > Thank you for the authors' response. Most of my concerns have been addressed, so I will raise the score. Some detailed data should be added to the paper to make it clearer for readers.

---

> > > ### Author Response · Authors · 2025-08-05
> > >
> > > Dear Reviewer rqDh,
> > >
> > > Thank you for your very encouraging and positive evaluation of our work. **We are especially grateful for your constructive suggestions and your willingness to consider raising the score based on our rebuttal**. We sincerely apologize for the initial omission you pointed out. **We will certainly incorporate the suggested details and corresponding data into the final manuscript** to enhance its clarity and readability.
> > >
> > > Thank you once again for your time and support.
> > >
> > > Best,
> > >
> > > Authors

---

### Note · Authors · 2025-08-12

Dear Area Chair,

We sincerely thank you for your dedication and support throughout the review process. To assist with your final assessment, we are writing to summarize the author-reviewer discussion for our paper **"LogicTree: …" (ID: 26975)**.

Our paper has received encouraging positive comments from the reviewers and we have diligently worked to address their remaining concerns. **We are sincerely grateful to the reviewers for their positive feedback and for their willingness to raise the score**.

Below is a summary of the key concerns and our corresponding resolutions:
## Reviewer rqDh
* **Computational Cost**: We added a detailed analysis about LogicTree's computational cost (measured in LLM output tokens).
* **Ablation Studies**: We conducted additional ablation experiments on the number of instantiated scenarios to validate our design choices.
* **Reliability of Verification**: We explained our error mitigation strategies during the validation process and performed a manual check to demonstrate the reliability.
## Reviewer sD1m
* **Dataset Statistics**:  We included additional detailed statistics for our synthetic dataset.
* **Correctness of Instantiation**: We clarified our two-stage instantiation method and systematic verification process to ensure correctness.
* **Experiments with Advanced LLMs**: We conducted additional experiments with more powerful models (e.g., DeepSeek-R1-Distill series).
## Reviewer xJSd
* **Algorithmic Complexity**:We provided a detailed analysis of the algorithmic complexity for LogicTree.
* **Experiments with Broader Models**: We conducted additional experiments with both advanced LLMs and smaller-scale models (e.g., 1.5B/3B) to demonstrate wider applicability.
* **Comparison with prompt-based methods**: We provided a detailed discussion about the key differences between them.
## Reviewer G8Cd
* **Core Contributions**: We further clarified the core contributions of our work for a more comprehensive understanding.
* **Conceptual Distinction**: We elaborated on the key differences between LogicTree and backward-chaining symbolic systems.
* **Key Details about LogicTree**: We provided point-by-point clarifications on each aspect to enhance clarity.

**We will incorporate these improvements into the final version to further strengthen our work**.

We hope this summary facilitates your review and thank you again for your time and consideration.

Best,

Authors

---

### Decision · Program_Chairs · 2025-09-17

**Decision:**

Accept (spotlight)

**Comment:**

The authors introduce “LogicTree”, a framework for synthesizing multi-step logical reasoning datasets with a focus on complexity and real-world contextual instantiation. The approach uses structural pattern matching for backward deduction to generate logic trees governed by first-order logic rules, and then employs a two-stage LLM-based method to instantiate these symbolic trees into diverse, context-rich, natural language scenarios. Logical consistency is checked post-hoc to filter poor samples. Experiments across several logical reasoning benchmarks demonstrate that training LLMs with LogicTree data improves accuracy over baselines.

Strengths of the paper:
- The logic tree generation based on structural pattern matching is well-grounded in formal logic.
- The experiments benchmark the method across diverse challenging datasets and tasks, including ablation studies for scenario instantiation and process diversity. Results support the main claims with quantitative improvements.

The reviewers have initial concerns about computation cost, reliability of verification, dataset stats, and some technical details. The author's responses properly addressed all of these concerns. All reviewers are satisfied with the responses.